# MXenes—A New Class of Two-Dimensional Materials: Structure, Properties and Potential Applications

**DOI:** 10.3390/nano11123412

**Published:** 2021-12-16

**Authors:** Maksym Pogorielov, Kateryna Smyrnova, Sergiy Kyrylenko, Oleksiy Gogotsi, Veronika Zahorodna, Alexander Pogrebnjak

**Affiliations:** 1Department of Nanoelectronics and Surface Modification, Faculty of Electronics and Information Technology, Sumy State University, 40007 Sumy, Ukraine; kateryna.v.smyrnova@gmail.com (K.S.); kyrylenk@gmail.com (S.K.); a.d.pogrebnjak@gmail.com (A.P.); 2Institute of Atomic Physics and Spectroscopy, University of Latvia, LV 1586 Riga, Latvia; 3Materials Research Centre, 03142 Kyiv, Ukraine; alex@mrc.org.ua (O.G.); veronika@mrc.org.ua (V.Z.); 4CARBON-UKRAINE Ltd., 03680 Kyiv, Ukraine; 5Department of Biotechnology, Faculty of Biology and Biotechnology, Al-Farabi Kazakh National University, Almaty 050040, Kazakhstan

**Keywords:** MXene, two-dimensional materials, carbides, properties, selective etching

## Abstract

A new class of two-dimensional nanomaterials, MXenes, which are carbides/nitrides/carbonitrides of transition and refractory metals, has been critically analyzed. Since the synthesis of the first family member in 2011 by Yury Gogotsi and colleagues, MXenes have quickly become attractive for a variety of research fields due to their exceptional properties. Despite the fact that this new family of 2D materials was discovered only about ten years ago, the number of scientific publications related to MXene almost doubles every year. Thus, in 2021 alone, more than 2000 papers are expected to be published, which indicates the relevance and prospects of MXenes. The current paper critically analyzes the structural features, properties, and methods of synthesis of MXenes based on recent available research data. We demonstrate the recent trends of MXene applications in various fields, such as environmental pollution removal and water desalination, energy storage and harvesting, quantum dots, sensors, electrodes, and optical devices. We focus on the most important medical applications: photo-thermal cancer therapy, diagnostics, and antibacterial treatment. The first results on obtaining and studying the structure of high-entropy MXenes are also presented.

## 1. Introduction

Studies of nanolayer two-dimensional materials date back to the 1950s [1,2]. Due to the discovery of graphene, it attracted the attention of many researchers for their methods of preparation, properties, and possible applications. Graphene has already demonstrated outstanding characteristics in terms of electrical conductivity, optical transparency, mechanical strength, and thermal conductivity [3,4]. Deep interest in graphene research facilitated the start of new two-dimensional solid materials, such as silicene, phosphorene, h-BN, and MoS_2_ [5,6,7,8]. These two-dimensional layered materials exhibit unusual electrical properties due to their unique characteristics, e.g., it is possible to place various ions or molecules between their layers [9].

In 2011, researchers from Drexel University discovered a new class of inorganic compounds with a two-dimensional structure, which aroused great interest among scientists from all over the world [10]. These unusual 2D materials were called MXenes. The family of MXenes consists of carbides, nitrides, or carbonitrides of transition metals. They are two-dimensional nanolayers with the thickness of several atoms and a planar size in the order of several micrometers [9,11,12,13]. MXene materials have a wide range of unique properties that make them attractive and suitable for a variety of applications [14,15,16]. For example, they have a large specific surface area and high mechanical, electronic, and physicochemical properties, as well as excellent biocompatibility [17,18].

To synthesize MXenes, MAX phases are used as the main precursor material. Their generalized formula is M_n+1_AX_n_, where M is an early transition metal, A is any element from IIIA or IVA (13–14) groups, X is carbon or nitrogen, but the combinations with the formation of carbonitride are also possible, and n = 1 − 4 [19]. Thus, the preceding M_n+1_AX_n_ phases have a hexagonal structure (P6_3_/mmc symmetry). In their simplest units, two constituent structural entities can be distinguished: octahedral "M_6_X" units with common edges, identical to the cubic face-centered lattice of NaCl of the type of binary nitrides or carbides, as well as the layers of the A element located between them. In this case, the X atoms are located in the space of octahedral interstices. It should be noted that the chemical bonds between the transition metal M and the X atom have a mixed character (metallic, ionic, and covalent components are present), while the M–A bonds are represented only by the metallic component. Therefore, the metal bond is weaker than the covalent one, and by choosing a suitable reagent for etching, it is possible to break the M–A bond and remove the elements of the A layer [20]. Consequently, selective etching of element A (for example, Al, Si, Ga, In, and S) leads to the formation of layered M_n+1_X_n_ with an "accordion-like" structure. Sometimes the formula M_n+1_X_n_T_x_ is used, where T is added to denote surface functional groups, for example, O, F, OH, H_2_O, and/or Cl [21]. The MXene layers obtained by etching the A element form various surface functional groups depending on the substance with which they react. Such reactions make it possible to control their physicochemical properties. For example, reacting with hydrofluoric acid (HF) and H_2_O, Ti_3_C_2_ acquires an F- and O/OH-terminated surface [22].

This review article provides an overview of the large family of 2D layered MXene materials. Particularly, studies on features of their structure and unusual properties, due to which they can be used for many various applications will be discussed. Furthermore, the paper also presents an analysis of precursor materials for the manufacture of two-dimensional MXenes and basic methods of synthesis and their modifications, such as selective etching and chemical vapor deposition. Finally, we will describe a number of useful applications of layered transition metal carbides in imaging and treatment of tumors, purification and desalination of water, as well as multifunctional smart textiles.

## 2. Methods to Synthesize MXenes

After successful synthesis of the first layered Ti_3_C_2_T_x_ using hydrofluoric acid, several other methods were developed and implemented to obtain various MXenes with new compositions. To date, there are two different strategies for obtaining MXenes using the “top-down” and “bottom-up” approaches (Figure 1). The main difference between these strategies is that the top-down synthesis method involves the separation of bulk materials, for example MAX phases, into several layers, while the bottom-up method uses the MXene deposition technology, forming the material from the bottom to the top. It should be noted that, depending on the synthesis method, the properties of the MXenes may differ. In addition, the characteristics of two-dimensional transition metal carbides/nitrides/carbonitrides depend on the starting material that is used to obtain them as well as various surface modification technologies. Therefore, a closer look is required to overview the main steps in the process of MXene synthesis and to identify the advantages and disadvantages of various strategies.

### 2.1. Materials for Synthesis of MXenes

#### 2.1.1. MAX Phases

The main materials used to obtain two-dimensional MXenes by selective etching are MAX phases. Their investigations began back in 1967. These materials were called *H* phases in [23,24], for instance, “211” phases (Cr_2_AlC) and “312” phases (Ti_3_SiC_2_). The pe-rounded by transition metal atoms. However, there was no bond between metametal atoms (Groups 13–14) and carbon (nitrogen). Since then, interest in the *H* phases faded away and resumed only 30 years later, when Barsum and his colleagues [25] obtained a predominantly single-phase bulk material Ti_3_SiC_2_, which possessed properties unusual for ceramics. Small bars of synthesized Ti_3_SiC_2_ were heated to 1400 °C and then quenched in water. They excellently withstood thermal shocks, exhibited high oxidation resistance, rigidity, thermal and electrical conductivity, and high melting points. Later, Barsum and his colleagues succeeded in synthesizing many new similar materials from the same family, for example, Ti_2_AlC, Ti_3_GeC_2_, Ti_4_AlN_3_, with, respectively, 211, 312, and 413 molecular compositions [26]. They called such materials MAX phases, and later, this name replaced the previous ones. The smallest units in the M_n+1_AX_n_ compounds have a hexagonal P6_3_/mmc space group with two formula units per unit. In this case, elements of the A layer in the 211, 312, 413, and 514 MAX phases are located in every third, fourth, fifth, and sixth layer, respectively [27]. The first MXene was synthesized from MAX phases by selective etching of Element A [10]. To this day, these compounds remain the main material for the preparation of two-dimensional carbides/sodium/carbonitrides of transition metals. Figure 2 shows all of the elements used to produce the MAX phases and MXenes and the functional surface groups in MXenes. In this case, the formation of simple MXene with one transition metal (M), A or X elements, and an ordered structure with two transition metals are possible, and the formation of a solid solution in the M, A, or X positions.

It was found that the addition of a new element to the composition of the MAX phases at the M or A positions can lead to their stabilization, which would otherwise be thermodynamically unstable [28]. Therefore, in 2008, new representatives of the MAX phase family appeared with the general formula (M_x_M′_1-x_)_n+1_AX_n_ (M and M′ are different transition metals), the M positions of which were a solid solution of two metals. A solid solution is formed by mixing elements located next to each other in the periodic table since they have similar electronic structures and atomic sizes [29]. The first such compounds were (V_0.5_Cr_0.5_)_2_AlC, (V_0.5_Cr_0.5_)_4_AlC_3_, and (V_0.5_Cr_0.5_)_5_Al_2_C_3_ [30], as well as (Cr_2/3_Ti_1/3_)_3_AlC_2_ and (Cr_5/8_Ti_3/8_)_4_AlC_3_ [31]. Such compounds allow to combine the advantages of Cr–Al–C and Ti–Al–C/V–Al–C systems, such as high-temperature resistance to corrosion and oxidation. Figure 3 shows Z-contrast images of the structure of (V_0.5_Cr_0.5_)_2_AlC, (V_0.5_Cr_0.5_)_4_AlC_3_, and (V_0.5_Cr_0.5_)_5_Al_2_C_3_ MAX phases, obtained using a scanning transmission electron microscope (STEM). In these images, light areas correspond to (V, Cr) layers and dark areas to Al. Please note that the atoms of vanadium and chromium cannot be distinguished due to the close atomic number.

As can be seen from Figure 3a, the atoms in (V_0.5_Cr_0.5_)_2_AlC are arranged similarly to the typical 211 MAX phase. Every two bright layers (V, Cr) are separated by a dark layer of aluminum atoms, known as ABABAB packing. The (V_0.5_Cr_0.5_)_4_AlC_3_ differs in that every three bright layers (V, Cr) are separated by one Al layer, which is described as ABCBCBABABC and is similar to the 312 phase. Figure 3c demonstrates the structure of (V_0.5_Cr_0.5_)_5_Al_2_C_3_, which is similar to the 523 phase and is a sequential stacking of two and three layers of carbides (V, Cr) with an Al layer located between each of them. It is a combination of halves of the elementary cells of the classical 211 and 312 phases [30]. Thus, it was discovered that it is possible to synthesize MAX phases with a solid solution in the M and A positions.

In 2014, chemically ordered MAX phases were discovered, for example, with the M_3_AX_2_ structure, where two elements M1 and M2 in a 2:1 ratio form a phase with a non-coplanar ("out-of-plane") chemical order through alternating layers consisting of one type M element. Liu et al. were the first to synthesize such a material. The result of the reaction between Cr_2_AlC and TiC at a molar ratio of Cr:Ti = 2:1 in M positions was an ordered 312 MAX phase (Cr_2/3_Ti_1/3_)_3_AlC_2_ (Figure 4) [32]. Unlike materials with a solid solution, in which some of the atoms are replaced by other atoms in the M or A positions in a chaotic manner, the ordered o-MAX phases exhibit an M-ordered crystal structure. Such "out-of-plane" ordered structures are called o-MAX phases. Soon, other representatives of this group were synthesized, for example, Mo_2_TiAlC_2_ [33], Mo_2_ScAlC_2_ [28] (312 phases) and Mo_2_Ti_2_AlC_3_ [34], Cr_2_V_2_AlC_3_ [35] (413 phases). A feature of the o-MAX phases is the presence of two different crystallographic M-sites with Wyckoff positions 4f for M_1_ and 2a for M_2_ in the case of 312, or 4f for M_1_ and 4e for M_2_ in 413 structures [35]. It was noted that for the formation of chemically ordered o-MAX, the metal layer M_1_, which is closest to the A layer, should not form a NaCl-type MC, and its electronegativity should be greater than that of the element of Layer A. The solid solution is observed in the case of small difference in the size and electronegativity of M_1_ and M_2_, combined with small differences in the electronegativity of M_1_ and A [36].

MXenes can be synthesized from the ordered MAX phases. Their atomic configuration differs in that the outer layers are occupied by M_1_ atoms, and the inner layers are composed of M_2_ atoms. For example, the first such two-dimensional carbides were Mo_2_TiC_2_T_x_ and Mo_2_Ti_2_C_3_T_x_, the structure of which is similar to "sandwiches", in which Mo atoms occupy the outer layers, while Ti atoms occupy the inner layers. This can be represented as follows: the outer Ti layers in MXene Ti_3_C_2_ and Ti_4_C_3_ are replaced by molybdenum atoms. These MXenes have not yet been thoroughly studied, but it is shown from [37,38] that, due to their structure, they exhibit unusual electrochemical characteristics. Thus, according to theoretical calculations, the discovery of these subfamilies could open up more than 25 new possible MXenes [39].

In addition to the o-MAX structures, one more group of these materials was studied. In 2017, Tao and colleagues [40] found that, in (Mo_2/3_Sc_1/3_)_2_AlC (M_2_AX), there was a different chemical ordering between two M elements; this time, M_1_ and M_2_ were ordered in "in-plane". To distinguish the new group of the ordered structures, they were named the i-MAX phase [36]. Unlike o-MAX phases, M_1_ and M_2_ in 211 structures are ordered in such a way that in each metal layer, they have a 2:1 ratio, that is, M_1_ atoms are located in a hexagonal lattice, and M_2_ atoms are located in the center of a hexagon (Figure 5a). i-MAX phases are formed in the case when the M_2_ atoms (minor metal) are larger than the M_1_ atoms (main metal), but electronegativity exhibits an inverse relationship (M_1_>M_2_) [41]. Such ordered materials exhibit a monoclinic structure with space group #15 C_2_/c or orthorhombic #63 Cmcm, while conventional MAX phases exhibit hexagonal #94 P6_3_/mmc [42,43]. The first representative of these materials, (Mo_2/3_Sc_1/3_)_2_AlC, was successfully used to synthesize Mo_1.33_C MXene. With the HF or LiF / HCl, not only the Al layer was removed but also selective etching of the M_2_ element (Sc). Interestingly, this process resulted in the formation of MXene with ordered divacancies [40].

With the discovery of i-MAX phases, in addition to the previously known methods of controlling the properties of MXenes (adding new elements or functionalizing the surface), a new approach was introduced, namely target etching [44]. It is known that the etching process of MAX-phases consists in removing weaker-bound elements in the starting material but retaining stronger bonds. For example, after opening the first 211 i-MAX phase, (W_2/3_Sc_1/3_)_2_AlC was received [43]. However, since Sc is a rare and expensive element, attempts have been made to replace it with other metals. This is how (V_2/3_Zr_1/3_)_2_AlC, (Mo_2/3_Y_1/3_)_2_AlC [41], (W_2/3_Y_1/3_)_2_AlC [43], and (Mo_2/3_RE_1/3_)_2_GaC (RE = Gd, Tb, Dy, Ho, Er, Tm, Yb, Lu) were synthesized [45]. It was found that using different etching protocols for the initial i-MAX phase (Mo_2/3_Y_1/3_)_2_AlC, it is possible to obtain two different MXenes: (1) (Mo_2/3_Y_1/3_)_2_C with chemical ordering in the plane (targeted removal of Al atoms) and (2) Mo_1,33_C with ordered vacancies (removal of Al and Y atoms) [29]. Figure 5b shows the process of obtaining two types of i-Mxene based on the i-MAX phase and shows the differences in their structure. To obtain Mxene, fine tuning of the protocol for selective etching of the starting material is required. For example, (Mo_2/3_Y_1/3_)_2_C was obtained using 48% HF for 12 hours, and for the synthesis of Mo_1,33_C, etching in 10% HF was chosen for 72 h [42].

Chemically ordered o-MAX and i-MAX phases are very attractive as precursors for Mxenes, since unconventional elements (Sc, Y, and W) can be added to them, and they allow better control of the alloy composition [46].

#### 2.1.2. Other Materials as Precursors of Mxenes

In addition to the MAX phases, it was found that other layered compounds can also be used as precursors for the synthesis of Mxenes [47]. Metals such as Sc, Zr, and Hf are more prone to forming layered ternary (MC)_n_Al_3_C_2_ or quaternary (MC)_n_[Al(A)]_4_C_3_ (n = 1, 2, 3) carbides [48]. Examples of such carbides are Zr_3_Al_3_C_5_, Zr_2_Al_3_C_4_, Hf_3_Al_3_C_5_, U_2_Al_3_C_4_, ScAl_3_C_3_, and Hf_3_[Al(Si)]_4_C_6_ [49]. The crystal structure of such compounds consists of alternative M-C sublayers of the NaCl type. It can be described as the structure of intergrowth of M-C layers of the NaCl type and Al_4_C_3_-like layers of Al_3_C_2_ or Al(A)]_4_C_3_ separating a carbon monolayer at the boundaries of their interaction. Figure 6 schematically shows a comparison of the structures of the MAX phase and two (MC)_n_[Al(A)]_m_C_m−1_ (m = 3 and 4) compounds. An analogy can be drawn between these structures if the atomic layer of Al in M_n+1_AlC_n_ for Al_3_C_3_ or [Al(A)]_4_C_4_. In this case, the layer of carbon atoms is common for the sublayers M-C and Al(A)-C at their common boundaries [49].

MC layers are chemically more stable than Al_4_C_3_, i.e. the latter is prone to hydrolysis in humid air or acidic environments. Therefore, in the case of Al–C, a weaker bond and increased reactivity are observed. By analogy with the MAX phases, the Al–c sublayers are preserved during etching, while the Al–C sublayers are broken [50]. For example, in [48,51], the process of obtaining Zr_3_C_2_T_x_ and Hf_3_C_2_T_x_ MXenes by the separation of Zr_3_Al_3_C_5_ and Hf_3_[Al(Si)]_4_C_6_ using concentrated hydrofluoric acid was described. Figure 7 shows the structure and schematic representation of the process of obtaining MXenes. The common atomic plane of carbon between the Hf–C and Al–C sublayers in the unit cell is also etched, resulting in Hf_3_C_2_T_x_. It was noted that the introduction of Si into the Al–C sublayer significantly accelerated the etching of layered ternary carbide due to the weakening of interfacial adhesion between the Hf–C and Al(Si)–C sublayers. Weakening of the Hf–C bond is explained by the fact that the atomic charge of silicon is higher (2.36) than that of aluminum (2.19). Thus, the common carbon layer is bound much stronger with the layer containing Al, which weakens the adhesive energy of the neighboring etching interface [51].

In addition to Al-containing carbides, layered Mo_2_Ga_2_C was successfully used to obtain MXenes in 2015 [27,52]. During the same year, Mo_2_C MXene was first obtained based on this material [53]. Mo_2_Ga_2_C was synthesized both in the form of a thin film and in the form of bulk material. This layered carbide is similar to the 211 MAX phase of Mo_2_GaC, to which an additional Ga layer has been added. In this case, the two layers of gallium are arranged in such a way that they lie symmetrically on top of each other. Figure 8 shows the unit cell of layered Mo_2_Ga_2_C. In its crystal structure, Mo and C atoms are located at positions 4f and 2a, and octahedra CMo_6_ with common edges are separated by two layers of Ga atoms. In this case, gallium atoms are weakly bound to CMo_6_ in the cubic structure [37].

Thus, it has been shown that MXenes can be synthesized using compounds other than the MAX phases. Examples of such materials are layered carbides with the formula (MC)_n_Al_3_C_2_, (MC)_n_[Al(A)]_4_C_3_ (n = 1, 2, 3), and Mo_2_Ga_2_C with a double rather than a single A atomic layer.

### 2.2. Strategies of MXene Synthesis

#### 2.2.1. Top-Down Approach

The most popular methods for obtaining MXenes are based on using agents to etch a specific layer from a bulk starting material. That is why such a process is called top-down, meaning that the synthesis takes place from top to bottom. The synthesis of two-dimensional MXenes from layered precursors (MAX phases or other materials) by selective etching of certain elements is possible due to the different activity of chemical bonds. For example, the M–A bonds are weaker compared to the M–X, which is why elements of the A layer can be removed without destroying the M–X bonds. The synthesis process of MXenes using the top-down methods can be divided into several steps (Figure 9). First, the precursor material is placed for a certain time in an etching reagent, designed to selectively remove elements of Layer A. After that, the treated material is sonicated to dissociate it into separate nanolayers in order to obtain ultra-thin two-dimensional MXenes. Two types of ultrasonic devices can be employed for this purpose: bath and probe/tip sonication systems. The probe sonicators produce smaller flake sizes of delaminated MXenes. At the same time, bath sonication has been found to be more suitable for fabricating MXenes with larger flakes. The most critical parameters of the sonication process are amplitude, power, and frequency [54]. The irradiation amplitude significantly influences the intensity of ultrasonication. The high vibration amplitudes during the treatment provide higher energy to the solution, which negatively affects the stability of MXene due to the severe collapse of cavitation bubbles, collisions between delaminated MXenes and particles of the MAX phase. Thus, it is necessary to choose an optimal amplitude value and lower the concentration of particles in solution through dilution. The amount of power should also be tailored: it must be sufficient to generate cavitation. In contrast to the amplitude, higher frequencies positively impact the MXene stability. They cause a high number of small bubbles with uniform sizes, preventing violent interparticle collapses and promoting the weakening of the interlayer Van der Waals forces [55]. Papers report different frequencies: from 6 kHz [56] to 40 kHz [57,58,59]. However, the highest quality of MXene flakes was observed at sonication frequencies above 20 kHz, and the most commonly used one is 40 kHz. The MXene synthesis process can be more efficient and faster by applying various intercalants, which can be organic molecules (dimethyl sulfoxide (DMSO), tetrabutylammonium hydroxide (TBAOH), tetrapropylammonium hydroxide), or inorganic materials (halide salts) [60].

The process of etching has a significant impact on possible applications of the resulting MXenes. For example, the widely used hydrofluoric acid (HF) is known to have a negative effect on biological organisms since even a small amount of HF that could remain after the stratification process of the starting material can induce toxicity for living cells. Therefore, HF is not generally used to produce MXenes for biomedical applications, but it can be used in other fields, e.g., catalysis, photo detectors, or energy storage systems. Due to these limitations, a number of other effective and safer agents to replace hydrofluoric acid have been developed, such as fluoride salts, ZnCl_2_, etc. In order to understand the advantages and disadvantages of different etchants, it is necessary to consider their features in detail.

##### Acid Etching

The first MXene synthesized back in 2011 was Ti_3_C_2_ [10]. For its preparation, about 10 g of Ti_3_AlC_2_ powder of the MAX-phase was used, immersed in 100 mL of a 50% concentrated HF solution at room temperature for 2 h. The resulting suspension was then washed several times with deionized water and centrifuged to separate the obtained MXene. Based on the studies carried out, it was found that, when Ti_3_AlC_2_ is immersed in an aqueous solution of hydrofluoric acid, several basic reactions occur, which can be written as follows [62]:(1)$${\mathrm{Ti}}_{3}{\mathrm{AlC}}_{2}+3\mathrm{HF}={\mathrm{AlF}}_{3}+\frac{3}{2}H_{2}+{\mathrm{Ti}}_{3}C_{2}\;,$$
(2)$${\mathrm{Ti}}_{3}C_{2}+2\mathrm{HF}={\mathrm{Ti}}_{3}C_{2}F_{2}+H_{2}\;,$$
(3)$${\mathrm{Ti}}_{3}C_{2}+2H_{2}O={\mathrm{Ti}}_{3}C_{2}{\left(\mathrm{OH}\right)}_{2}+H_{2}\;,$$
(4)$${\mathrm{Ti}}_{3}C_{2}+O_{2}={\mathrm{Ti}}_{3}C_{2}O_{2}\;$$

Thus, the primary reaction (1), which describes the process of separation of the MAX phase with the formation of MXene, AlF_3_, and H_2_ (during the course of the chemical reaction, the formation of bubbles was observed), was accompanied by processes (2–4) responsible for the formation of O, OH, and F surface groups. Later, it was found that after etching in a 50% HF solution, the synthesized Ti_3_C_2_ contained endings that were a mixture of -F and -O/OH.

After the first successful use of hydrofluoric acid for the preparation of MXene, it became the most popular reagent used for selective etching [10,48]. Only the concentration of HF in an aqueous solution (5–50 wt.%) varied, as well as the immersion time of the MAX phase and different precursor materials were employed. However, with an increase in the HF concentration, an increase in the number of defects in the synthesized MXene was observed. For comparison, Figure 10 shows X-ray diffraction patterns and micrographs of Ti_3_AlC_2_ and Ti_3_C_2_T_x_ synthesized at various HF concentrations and reaction times. Three MXenes were obtained at 30, 10, and 5 wt.% HF for 5, 18, and 24 h, respectively [63]. Using a higher concentration of hydrofluoric acid could shorten the reaction time. Based on the analysis of X-ray spectra, it can be concluded that the chosen etching parameters were sufficient to completely remove Layer A from the MAX phase. Scanning electron microscopy (SEM) images clearly show that all the resulting MXene powders have a layered structure, while their appearance was slightly different depending on the acid concentration. In the case of 30% HF, the morphology of multilayer Ti_3_C_2_T_x_ was the most similar to an accordion-like shape with large gaps between the layers. However, for lower acid concentrations, such a morphology was not observed and was more similar to the MAX phase. The increase in the distance between the layers of MXene synthesized at 30 wt.% HF could occur due to the large amount of gaseous H_2_ evolved during the reaction. Despite the external differences, in all three cases, MXene contained no Al impurities.

The difference between the structures of the Ti_x_Ta_(4−x)_AlC_3_ MAX phase and the Ti_x_Ta_(4−x)_C_3_ MXene obtained using 40% solution of hydrofluoric acid is shown in Figure 11 [64]. The images obtained with transmission electron microscopy (TEM) clearly show the layered structure of these materials. Darker areas are seen due to overlapping layers. It can also be seen that the interplanar distances, d, of Ti_x_Ta_(4−x)_AlC_3_ and Ti_x_Ta_(4−x)_C_3_ are different. Due to the removal of the aluminum layer from the MAX-phase structure, an increase in the d-distance of the synthesized MXene is observed. The figure also shows that the MXene surface is rougher than the MAX phase. This occurs as a result of the formation of functional surface groups on the surface during etching. 

Consequently, the procedure for preparing MXene using concentrated HF solution simply follows a few basic steps. First, the precursor material was added to the hydrofluoric acid solution. This was undertaken gradually in order to reduce the number of bubbles released during the exothermic reaction of HF with Al [63]. As a result, a multi-layered MXene powder was obtained, while the two-dimensional layers were held by Van der Waals forces and hydrogen bonds. Reaction by-products, unreacted precursor powder, and residual HF were removed by centrifugal washing in deionized water several times. After each centrifugation cycle, sediment at the bottom of the tube was the MXene powder. The supernatant (liquid above the solid precipitate) was poured out, and deionized water was again added to the test tube. This was repeated several times until the supernatant with a pH of about 6 was achieved. MXenes were then collected by vacuum filtration and dried [65].

Another method for obtaining Mo_2_C MXene was described by etching Mo_2_Ga_2_C with H_3_PO_4_ using ultraviolet (UV) radiation was used to accelerate the process [37]. The mixture was stirred in a magnetic stirrer under irradiation with UV light (100 W, working distance ~8 cm) for 3–5 h. Then, several centrifugation steps were performed to wash the MXenes with water until the pH of the residual solution raised to about 7, followed by dispersion using ultrasonic treatment in bath type sonication system in oxygen-free deionized water, and finally, the samples were vacuum dried. As a result, mesoporous Mo_2_C was obtained with a high degree of purity and unique properties for usage in rechargeable batteries and flexible batteries. The layered structure of this Mo_2_C MXene and pores with a diameter of 2–16 nm are shown in Figure 12. Such MXene contains a small amount of O functional groups on the surface. UV etching gives good potential as it allows avoiding the use of hazardous F-containing acids.

##### Etching with Fluoride Salts

Using HF poses substantial risks due to the fact that it can penetrate through the skin and other body tissues to cause the damage. Therefore, other methods to replace HF have been investigated [66]. One such method proposes to use inexpensive hydrochloric acid (HCl) and fluoride salts, which leads to the formation of HF in situ [67]. This method has become widely used for the synthesis of MXenes. LiF, NaF, KF, NH_4_F, and CaF_2_ are employed as fluoride salts [61]. Such etchants have been found to be much milder than hydrofluoric acid. This is manifested by the formation of MXene flakes with large transverse dimensions, which do not contain nanoscale defects as with HF [68].

Using such etchants, the first MXene was obtained in the form of a clay-like paste [67]. In this case, LiF was dissolved in 6M HCl, and after the slow addition of the Ti_3_AlC_2_ MAX phase, the mixture was warmed up to 40 °C and incubated for 45 h. Similar to the selective etching procedure using concentrated HF to raise the pH and remove the reaction byproducts, the mixture was washed several times in deionized water followed by centrifugation and decanted. No intercalation, layering, or filtration was used in the case of etching with the LiF/HCl mixture. The resulting precipitate was passed in a wet state between permeable membranes in a roller mill. The end product of this procedure was a clay-like paste, with which flexible free-standing films could be produced. Such a paste swells when hydrated and shrinks when dried. This is possible due to the introduction of layers of water and, possibly, cations between the hydrophilic and negatively charged Ti_3_C_2_ MXene sheets, which was manifested by a significant increase in the crystal lattice parameter c. Unlike weaker etchants based on fluoride salts, hydrofluoric acid does not allow synthesizing MXenes, which can be folded and hydrated.

In 2016, Wang et al. [66] presented a new method for synthesizing MXene by hydrothermal MAX phase separation using NH_4_F. For selective etching of Ti_3_AlC_2_, 5 g of NH_4_F dissolved in 60 ml of deionized water was used and vigorously stirred with a magnetic stirrer. The reaction mixture was then placed at 150 °C for 24 h in a Teflon-lined stainless-steel autoclave. Then, it was cooled naturally, centrifuged several times, and washed with warm deionized water (40–50 °C) and absolute ethyl alcohol. Finally, it was vacuum dried at 60 °C for 12 h. The morphology and elemental composition of Ti_3_C_2_ MXene synthesized by the hydrothermal method using NH_4_F are shown in Figure 13. The SEM image indicates the complete separation of the MAX phase, but nanoparticles are visible on the surface of the layers. By energy dispersive analysis, Ti, C, O, F, Al, and N were found, and their concentrations were as follows: 33.4, 20.47, 31.56, 13.11, 0.98, and 0.48 at.%. The ratio of titanium to carbon was 3:2. The obtained MXene contained a high concentration of O due to large amount of OH functional groups on the surface and the formation of TiO_2_ due to the reaction carried out at the elevated temperature. A small amount of N could be explained by one of the reaction products (NH_4_)_3_AlF_6_ and a possible residual amount of NH_4_F. Thus, the equations describing the reaction between NH_4_F and the MAX phase with the formation of Ti_3_C_2_T_x_ MXene can be written as follows:(5)$${\mathrm{NH}}_{4}F+H_{2}O↔{\mathrm{NH}}_{3}\cdot H_{2}O+\mathrm{HF}$$
(6)$${\mathrm{Ti}}_{3}{\mathrm{AlC}}_{2}+3\mathrm{HF}={\mathrm{Ti}}_{3}C_{2}T_{x}+{\mathrm{AlF}}_{3}+\frac{3}{2}H_{2}$$
(7)$${\mathrm{NH}}_{4}F+{\mathrm{AlF}}_{3}={\left({\mathrm{NH}}_{4}\right)}_{3}{\mathrm{AlF}}_{6}$$

One of the reaction products of NH_4_F and water is HF, after which HF reacts with Ti_3_AlC_2_ to form MXene, AlF_3,_ and the evolution of gaseous H_2_. In turn, due to the interaction of NH_4_F with AlF_3_, (NH_4_)_3_AlF_6_ is formed. The disadvantage of this method is the formation of titanium oxides on the surface of the synthesized MXene.

In 2017, the first nitride MXene Ti_2_NT_x_ was obtained by selective etching of aluminum atoms from Ti_2_AlN using a mixture of KF and HCl for 3 h, followed by heating for 1 h at 40 °C and sonication [69]. The synthesized multilayer MXene was then delaminated with dimethyl sulfoxide. It was previously believed that the use of fluoride salts at low temperatures did not allow the MAX phase to be transformed into two-dimensional MXene. Unlike HF, the KF–HCl etchant simultaneously promotes the intercalation of K^+^ ions, which, in turn, weakens the M–X bonds. Thus, using a mixture of potassium fluoride and hydrochloric acid forms a mixture of fluoride and chloride salts. The equations of possible reactions can be represented as follows:(8)$$2{\mathrm{Ti}}_{2}\mathrm{AlN}+6\mathrm{KF}+6\mathrm{HCl}=2{\mathrm{Ti}}_{2}{\mathrm{NT}}_{x}+K_{3}{\mathrm{AlF}}_{6}+{\mathrm{AlCl}}_{3}+3H_{2}+3\mathrm{KCl}\;$$
(9)$$2{\mathrm{Ti}}_{2}\mathrm{AlN}+6\mathrm{KF}+6\mathrm{HCl}=2{\mathrm{Ti}}_{2}{\mathrm{NT}}_{x}+2{\mathrm{AlF}}_{3}+3H_{2}+6\mathrm{KCl}\;$$

Later, a new V_2_NT_x_ MXene (T = F or O) was obtained, while a LiF–HCl mixture was used as the etchant. The layering of two-dimensional multilayer vanadium nitrides was carried out using a DMSO solvent and manual shaking [70].

##### Etching with Amonium Hydrofluoride

Alhough the use of HCl with fluoride salts reduces the use of hydrofluoric acid, the reaction still produces a volatile HF gas that can harm the environment [71]. Therefore, in 2014, Halim and his colleagues [62] proposed using ammonium hydrofluoride (NH_4_HF_2_) as an etchant. To synthesize Ti_3_C_2_ MXene, they used thin Ti_3_AlC_2_ films obtained by DC magnetron sputtering on an (0001) Al_2_O_3_ substrate. During the deposition of the precursor for MXene, an intermediate TiC (111) layer 5–10 nm thick was formed, which promoted the growth of epitaxial Ti_3_AlC_2_. Two different reagents, 50% HF and 1M NH_4_HF_2,_ were used to synthesize two-dimensional carbides based on a 60 nm thick film. The time required for the selective etching of Al with a bifluoride etchant was much longer (660 min) than the reaction time with concentrated hydrofluoric acid (160 min). Figure 14 shows the X-ray diffraction patterns of the deposited Ti_3_AlC_2_ film and two Ti_3_C_2_T_x_, as well as the results of X-ray photoelectron spectroscopy (XPS). Analysis of X-ray diffraction patterns and XPS spectra showed that with NH_4_HF_2_, like with HF, it was possible to successfully synthesize MXene. For the Ti_3_AlC_2_ precursor, the lattice parameter (c) was about 18.6 Å. However, Ti_3_C_2_T_x_-HF showed a shift of 000l diffraction peaks towards smaller angles, corresponding to an increase in c to 19.8 Å. Moreover, in the case of using NH_4_HF_2_, the crystal lattice of Ti_3_C_2_T_x_ expanded to 24.7 Å. The increase in the lattice parameter occurs due to the selective etching of Al atoms. The high-resolution XPS spectrum in the N 1s region (Figure 14e) obtained for the NH_4_HF_2_ MXene shows peaks corresponding to NH_4_^+1^ and NH_3_. Thus, the presence of intercalated cations causes a significant 33% increase in the lattice parameter c. It can be concluded that the use of ammonium hydrofluoride leads to simultaneous etching and intercalation of NH_3_ and NH4^+^ cations, which facilitates the procedure for obtaining two-dimensional MXenes. The reaction of the interaction of the MAX phase (Ti_3_AlC_2_) and NH_4_HF_2_ is as follows:(10)$${\mathrm{Ti}}_{3}{\mathrm{AlC}}_{2}+3{NH;}_{4}{\mathrm{HF}}_{2}={\left({NH;}_{4}\right)}_{3}{\mathrm{AlF}}_{6}+\frac{3}{2}H_{2}+{\mathrm{Ti}}_{3}C_{2}$$
(11)$${\mathrm{Ti}}_{3}C_{2}+{\mathrm{aNH}}_{4}{\mathrm{HF}}_{2}+{\mathrm{bH}}_{2}O={\left({\mathrm{NH}}_{3}\right)}_{c}{\left({\mathrm{NH}}_{4}\right)}_{d}{\mathrm{Ti}}_{3}C_{2}{\left(\mathrm{OH}\right)}_{x}F_{y}$$

Equation (7) shows that the etchant NH_4_HF_2_ differs from HF in that, instead of AlF_3_, (NH_4_)_3_AlF_6_ is formed as a result of the reaction. Moreover, in addition to the intercalation of NH_3_ and NH4^+^, the formation of terminal hydroxyl and fluoride groups on the MXene surface is also observed (8).

In 2020, Natu and colleagues [72] proposed a method for selectively etching a precursor material without using water as the main solvent. This approach broadened the range of MXene applications. They synthesized Ti_3_C_2_ MXene using polar organic solvents and ammonium hydrogen fluoride. This method was developed based on the hypothesis that NH_4_HF_2_ dissociates into NH_4_F and HF when dissolved in polar solvents. For etching the phase with 1 g of Ti_3_AlC_2_ MAX phase, 10 ml of propylene carbonate (PC) and 1 g of dehydrated NH_4_HF_2_ were used. The resulting mixture was stirred inside the glove box for 196 h at 35 °C, then washed five times using a 6M HCl solution in 2-propanol, and then the tube with the mixture was centrifuged several times with the addition of propylene carbonate. To separate Ti_3_C_2_T_x_ into one or several layers, the mixture with propylene carbonate was sonicated in the presence of Ar for an hour, immersed in an ice bath, and centrifuged again to separate non-delaminated MXene, salts, and the remaining MAX phase particles. This method allows the entire synthesis to be carried out in a glove box if necessary. As a result of this procedure, a highly fluorinated MXene was obtained. Electrodes made from two-dimensional Ti_3_C_2_T_x_ using propylene carbonate demonstrated almost twice the capacity when tested as anodes in a Na-ion battery [72].

##### Reaction with Alkali

Both free HF and in situ HF have been shown to be effective for the synthesis of MXenes. However, hydrofluoric acid negatively effects the environment and the human body and also leads to formation of MXenes with inert F functional surface groups, thus reducing some of the characteristics of the material, such as capacity. Therefore, Li et al. [73] in 2018 proposed a new method for selective etching of the Ti_3_AlC_2_ precursor material using a hydrothermal process using NaOH. The use of alkali as an etchant made it possible to obtain Ti_3_C_2_T_x_ with terminal OH and O groups without fluorine.

Figure 15 schematically shows the process of selective etching of Al using an aqueous solution of NaOH. The impact of OH- on Al in the MAX phase during the reaction occurs in two stages. First, aluminum is oxidized to Al hydroxide (oxide), and then the latter is dissolved in alkali in the form of soluble Al(OH)_4_^−^. With continued etching, the internal Al atoms begin to get further oxidized. Bayer demonstrated that higher temperatures and higher concentrations of alkalis promoted the dissolution of aluminum hydroxides (oxides). Therefore, for the successful preparation of MXenes by selective etching of the MAX phase, in addition to a high concentration of NaOH, the reaction must proceed at elevated temperatures. Thus, Ti_3_AlC_2_ was added to a 27.5 M NaOH aqueous solution at 270 °C, resulting in a Ti_3_C_2_T_x_ powder of high purity (about 92 wt.%). The synthesized MXene had -OH/-O end groups on the surface, and partial intercalation of Na^+^ into interlamellar positions was observed as a result of treatment with a NaOH solution. To prevent the oxidation of titanium, the hydrothermal treatment was carried out in an argon atmosphere. MXene synthesized with NaOH showed a gravimetric capacity superior to the analogs obtained by HF etching.

##### Reaction with Molten Fluoride Salts

An aqueous solution of hydrofluoric acid, which is widely used to synthesize carbide and carbonitride MXenes, is not suitable for the selective etching of nitride MAX phases. It was found that replacing X atoms from C to N radically changes the behavior of the MAX phase during etching. In the nitride precursor, the A atoms of the layer are stronger with the Ti–N block, which prevents the formation of the TiN_x_ phase upon reaction with HF [74]. Therefore, the first two-dimensional nitride MXene, Ti_4_N_3_T_x_, was synthesized in 2016 [75]. For this synthesis, a new method was used based on heating the MAX phase in molten fluoride salts. Figure 16 shows a scheme of the protocol for the synthesis of nitride MXene and images of the Ti_4_N_3_T_x_ surface before and after delamination. The Ti_4_AlN_3_ powder of the MAX phase was used as a precursor. The etchant consisted of a mixture of fluoride salts in the following concentrations: 59 wt.% KF, 29 wt.% LiF, and 12 wt.% NaF. The reaction was carried out at 550 °C for 30 min in an Ar atmosphere. Etching products (Al-containing fluorides) were dissolved with diluted H_2_SO_4_ and removed using deionized water, followed by repeated centrifugation and decanting the supernatant until it reached the pH of about 6. The last step was filtration of the precipitate through a polypropylene membrane. Delamination of the multi-layer MXene and the removal of un-etched MAX phase particles was carried out using tetrabutylammonium hydroxide (TBAOH) and deionized water. As a result of etching with a mixture of molten MXene salts, as in the case of HF, a surface with F-, O-, and OH-terminal groups was obtained. The formation of titanium oxide was also observed, which was most likely associated with the slow oxidation of nitride MXene in the aqueous solution.

Another example of the use of molten salts for the synthesis of MXene was presented in [76]. The authors obtained Ti_3_C_2_Cl_2_ using the Ti_3_AlC_2_ MAX phase and ZnCl_2_ in a molar ratio of 1:6 after heat treatment at 550 °C. The formation of MXene with a Cl-terminated surface proceeded as follows: Ti_3_AlC_2_ → Ti_3_ZnC_2_ → Ti_3_C_2_Cl_2_. The sample that was heat-treated for 30 min consisted of Ti_3_AlC_2_, but annealing for 60 min caused the transformation of the MAX phase with the formation of Ti_3_ZnC_2_. After 90 min of the reaction, Ti_3_C_2_Cl_2_ MXene with Zn impurities was formed.

At 280 °C, ZnCl_2_ melts and ionizes to Zn^2+^, which is a strong acceptor of Cl^-^ and electrons. As a result, an acidic environment is formed, in which weakly bound Al atoms in the MAX phase are converted to Al^3+^ using a redox reaction. In turn, the latter reacts with Cl^-^ to form AlCl^3^ (boiling point about 180 °C), which evaporates at 550 °C. The vacancies formed after the removal of Al are filled with reduced zinc atoms, which leads to the formation of Ti_3_ZnC_2_. After 30 min in ZnCl_2_ at 550 °C, Ti_3_ZnC_2_ forms Ti_3_C_2_Cl_2_ MXene. The following equation sumarizes this process:(12)$${\mathrm{Ti}}_{3}{\mathrm{ZnC}}_{2}+{\mathrm{ZnCl}}_{2}={\mathrm{Ti}}_{3}C_{2}{\mathrm{Cl}}_{2}+2\mathrm{Zn}$$

Weakly bound zinc atoms in Ti_3_ZnC_2_ are easily removed from the A-sites and dissolve in the ZnCl_2_ salt melt. In turn, intercalation of the Cl^−^ anions occurs, followed by the formation of electrons and the more stable Ti_3_C_2_Cl_2_ phase. As a result of the capture of electrons by zinc ions, pure Zn is formed. Thus, Zn^2+^ and Cl^-^ in this method, as well as H^+^ and F^-^ in the etching method with hydrofluoric acid, play similar roles. However, the preparation of a stable colloidal solution with one or more layers of Cl-terminated MXene has not yet been reported. One of the reasons may be the absence of O/OH functional groups, which are associated with colloidal stability after the separation process [72].

##### In-Situ Electrochemical Synthesis

In 2020, a new one-step protocol was presented to manufacture energy storage devices with MXene [77]. The authors proposed to combine two processes, namely the synthesis of MXene and the subsequent multistage assembly of devices. This method is also different because it does not use acidic or alkaline etchants, and all reactions proceed without contact with the external environment. The efficiency of a one-step protocol for the synthesis and manufacture of a zinc-based water battery has been demonstrated with the V_2_CT_x_ MXene as an example. A closed button cell CR2030 was created, which consisted of a V_2_AlC MAX phase (cathode), Zn (anode), and 21 M LiTFSI + 1 M Zn (OTf)_2_ (high fluorine electrolyte for the electrochemical process). That is, V_2_AlC exfoliated directly inside the battery, while synthesized in situ V_2_CT_x_ with transverse dimensions of about 1–5 μm and a thickness of about 8.5 nm (5–7 layers) maintained its high characteristics. As in the case of using the HF solution, selective etching of aluminum atoms occurs, and F and O/OH functional groups are formed on the surface of the obtained MXene. The reaction equation for the transformation of the precursor in MXene can be represented as follows:(13)$$V_{2}\mathrm{AlC}+{\gamma F}^{-1}+\left(2x+z\right)H_{2}O-\left(\gamma +3\right)e^{-1}=V_{2}C{\left(\mathrm{OH}\right)}_{2x}F_{y}O_{z}+{\mathrm{Al}}^{3+}\left({\mathrm{Al}}_{2}O_{3},{\mathrm{AlF}}_{3}\right)$$

Thus, V_2_C MXene is formed as a result of breaking the MA bonds in the MAX phase by the influence of F^−1^ from OTf^−1^. A decrease in the surface energy of MXene occurs through the formation of functional groups as a result of the reaction with H_2_O and F^−1^. Then, one more stage of the device operation was observed, namely oxidation of V_2_CT_X_ MXene to V_2_O_5_. The authors described this reaction by the following equation:(14)$$V_{2}{\mathrm{CT}}_{x}+5H_{2}O=V_{2}O_{5}+C+{\mathrm{xF}}^{-}+{\mathrm{yOH}}^{-}$$

The V_2_O_5_ reaction product improves productivity and increases capacity. The operation of such a zinc ion battery involves the separation of the MAX phase, the oxidation of the electrode, and the redox reaction without contact with the environment in a closed cell. It is an environmentally friendly and fast method of making MXene-based energy storage devices.

#### 2.2.2. Bottom-Up Approach

##### Chemical Vapor Deposition

In addition to the widely used top-down methods for preparing MXenes, based on the separation of the precursor material using an etchant, a bottom-up approach for the synthesis of two-dimensional carbides has also been developed [69]. The bottom-up strategy relies on the formation of MXenes via controlled deposition of multilayer epitaxial films. In contrast to the top-down methods, which often require ultrasonic treatment leading to a decrease in MXene size and formation of defects, this approach allows the synthesis of two-dimensional carbides with large transverse dimensions and significantly fewer structural defects [78].

The first epitaxial Ti_3_C_2_ thin film was synthesized by Halim and colleagues [62]. The MAX phase in the form of a film deposited by DC magnetron sputtering in an ultrahigh vacuum was used as a precursor. However, transparent films of the Ti_3_C_2_T_x_ MXene were still produced by selectively etching the A layer with ammonium hydrofluoride.

Later in 2015, Xu et al., for the first time, developed the process of synthesis of ultrafine and defect-free α-Mo_2_C without using any etchants [77]. Crystals of α-Mo_2_C with a thickness of not more than 3 nm and a transverse size of more than 100 μm were obtained by chemical vapor deposition for 5 min at a temperature above 1085 °C atatmospheric pressure. A Cu 10 μm thick foil was placed on top of a Mo foil as a substrate and then heated to a temperature above the melting point of copper in hydrogen so that molybdenum atoms diffused to the surface of liquid copper. Then, CH_4_ was introduced, which was used as a source of C, which can react with Mo on the copper surface with the subsequent formation of ultrafine α-Mo_2_C. Figure 17a schematically shows the synthesis process of two-dimensional crystals of molybdenum carbide and models of their atomic structure depending on thickness. The resulting crystals had regular shapes, such as triangles, rectangles, hexagons, and octagons (Figure 17b). The electron diffraction pattern of the selected region showed that, as a result of synthesis by chemical vapor deposition, the formation of Mo_2_C crystals with an orthorhombic structure was observed. That is, the molybdenum atoms were in a slightly distorted hexagonal close-packed arrangement, and the C atoms occupied ordered positions in the octahedral voids of the Mo lattice (Figure 17c–f). Such ultrathin α-Mo_2_C crystals had superconductivity, which depended on the crystal thickness, and strong anisotropy with the orientation of the magnetic field [65].

A year later, single crystals of Mo_2_C with controlled thickness and morphology were synthesized on the surface of liquid copper by a similar chemical vapor deposition method under atmospheric pressure but at temperatures above 1100 °C [79]. It has been shown that the thickness of the films can be precisely controlled by varying the thickness of the copper foil catalyst. Moreover, the morphology of crystals can be controlled by adjusting the concentration of carbon. 

Later, Wang et al. [80] presented a new method for synthesizing Mo_2_C in the form of thin films for using as a broadband nonlinear optical medium for optical applications. They used a Mo_2_C target for deposition, which was sputtered onto a quartz wafer substrate by radio frequency magnetron deposition at a pressure of 6 × 10^–4^ Pa. Then, the synthesized Mo_2_C samples were annealed at a temperature of 600 °C. The MXene was approximately 4.3–4.7 nm thick. X-ray photoelectron spectroscopy showed that Mo_2_C also had Mo–O and C–O bonds, for which the oxidation products of the target could be responsible. The deposition of a 20 nm thick SiO_2_ thin film on MXene made it possible for the first time to obtain a saturable absorber for passively Q-switched solid-state lasers doped with neodymium (Nd:YAG and Nd:YVO_4_). 

Bottom-up materials are of high crystal quality without defects. However, it is challenging to synthesize large-area atomically thin films. The development of these approaches will make it possible to obtain new two-dimensional transition metal carbides and nitrides with interesting characteristics for various applications, for example, energy storage systems and catalysis [78].

### 2.3. Delamination of MXenes Using Intercalating Agents

After selective etching of the A layer of the precursor material, multilayer MXenes are formed. In order to speed up and facilitate the process of obtaining separate two-dimensional layers, various intercalating compounds are widely used. As a result of their intercalation, an increase in the distance between the layers is observed in the accordion-like structure formed after the etching of the precursor. This significantly weakens the strong interlayer interactions in multilayer MXenes, which are difficult to break by simple mechanical exfoliation [63]. Figure 18a schematically shows the top-down synthesis of MXene using an intercalant.

Intercalation substantially speeds up the separation process of MXenes. After the addition of the intercalant material, the MXene multilayer solution is mixed and then washed and centrifuged to separate the exfoliated material until the pH of the supernatant gets stabilized at ~7. Sometimes, after mixing, the sonication stage is additionally employed. Prolonged sonication at high power, however, leads to a decrease in the planar dimensions of the MXene, as well as to an increased number of defects [63]. The use of intercalants allows obtaining single layer MXenes, resistant to aggregation and adhesion [54]. There are two types of intercalants: organic solvents and metal cations [81].

Representatives of organic intercalants are dimethyl sulfoxide (DMSO), isopropylamine, urea, hydrazine monohydrate (N_2_H_4_ · H_2_O, HM), tetrabutylammonium hydroxide ((C_4_H_4_)_4_NOH), TBAOH), tetramethylammonium hydroxide (CH_3_)_4_NOH, TMAOH), *n*- butylamine (C_4_H_11_N), and choline hydroxide ((C_5_H_14_NO)OH) [81,82]. For example, Mashtalir et al. [9] used hydrazine monohydrate dissolved in N,N-dimethylformamide (DMF) for intercalation. Figure 18b,c shows that, after intercalation, MXene remains delaminated and thickening of multilayers is observed, apparently due to gluing individual monolayers together, forming lamellas. Subsequent sonication and centrifugation provided two-dimensional single MXene nanolayers. When using DMSO, dispersing MXene sheets in solution requires sonication to prevent the multilayer carbides from settling as a sludge. In this case, nanolayers exfoliated with DMSO usually have dimensions on the order of several hundred nm [63,83]. Tetraalkylammonium compounds (TBAOH and TMAOH) can also be successfully used as intercalants [84,85]. Unlike DMSO, which involves the use of ultrasonic treatment, TBAOH and TMAOH are based on ion exchange between protons and bulk tetraalkylammonium ions, TMA^+^ or TBA^+^. Consequently, the MXene layers "swell", which leads to delamination even with manual shaking [65,82]. For example, a positive effect of TBA^+^ intercalation on the electronic properties (increased sample resistance and negative dR/dT) of MXene was observed, even after annealing above 750 °C. These results are relevant for the optimization of MXene devices, where high metallic conductivity is required, and the ability to control dR/dT through intercalation can be used for sensors [84].

The second type of intercalants is metal ions such as halide salts or metal hydroxides in aqueous solutions. Various metal cations such as NH^4+^, Mg^2+^, Al^3+^, Ca^2+^, Li^+^, Na^+^, K^+^, and Sn^4+^ can spontaneously intercalate due to the negative charge of MXenes [86]. After intercalating between nanolayers, such ions increase the distance between them, thereby weakening the Van der Waals force, which is responsible for forming the multilayer structure of MXene. Repeated washing of the slurry in deionized water subsequently raises the pH and gradually swells the multilayer MXene. After that, two-dimensional monolayer carbides can be obtained by manual shaking or sonication [65]. For example, it has been demonstrated that the use of MXene with intercalated K^+^ cations for the NF electrode led to an increase in the specific capacity by more than two times compared to conventional MXene, and also contributed to significantly higher efficiency of ion and electron transfer [87]. It was also found that the presence of Li^+^ cations in Ti_3_C_2_T_x_ dramatically increases the electrical conductivity of MXene [67]. Li^+^ intercalation is accompanied by spontaneous intercalation of water molecules between layers of two-dimensional carbides. The use of other cations for intercalation, such as Na^+^, K^+^, Ca^2+^, and Mg^2+^, also positively affected the characteristics of Ti_3_C_2_T_x_. They modulated the structural response to relative humidity and hydration [88].

## 3. Structure and Properties

MXenes are unique because they have many properties attractive for various applications, such as hydrophilicity, high thermal and electrical conductivity, chemical stability, excellent biocompatibilitym etc. These properties make MXenes ideally suited in areas such as electromagnetic shielding interference, wireless communication, chemical sensors, energy storage systems, optoelectronics, catalysis, and flexible electronics. Table 1 shows various MXenes, their properties and applications.

### 3.1. Structure

A typical MXene structure is shown schematically in Figure 19 for M_3_X_2_ derived from an M_3_AX_2_ precursor. It can be considered a layer extracted from the fcc titanium carbide lattice along the (111) plane. The atoms of the A layer are dissolved by etching, leaving the highly active M atoms of the transition metal on the surface of the nanolayers with X atoms inside. Figure 19d schematically shows the location on the surface of the end groups that stabilize the surface atoms of the transition metal [100]. The synthesized MXene has a close-packed hexagonal structure. Depending on n, the metal atoms have a different ordering. For n = 1, an ABABAB stacking sequence is observed, while MXene with n ≥ 2 exhibits ABCABC ordering.

Theoretical calculations predict the possibility to obtain about 100 stoichiometric structures, from which approximately 20 different compositions have been synthesized experimentally [21]. Some of the most studied materials of this class are Ti_3_C_2_, Ti_2_N, Nb_4_C_3_, Nb_2_C, and V_4_C_3_ [62]. Figure 20 shows images of Ti_3_C_2_, the first representative of a new class of two-dimensional materials, obtained using a scanning electron microscope and a scanning transmission microscope. Figure 20a clearly shows the accordion-like layered structure of MXene. The voids between the Ti_3_C_2_ layers were formed as a result of removing the Al (A layer) from the parent Ti_3_AlC_2_ MAX phase. It can be seen that the layers of nanosheets are glued in some places, which occurs under the action of Van der Waals forces and hydrogen bonds. So far, four different configurations of MXene structure as shown in Figure 21 have been synthesized. 

Additionally, as mentioned above, there are out-of-plane ((M′_2_, M′′)AlC_2_ or (M′_2_, M′′_2_)AlC_3_) and in-plane ((M′_2/3_, M′′_1/3_)AlC) MAX phases, which can be etched to synthesize the o-MXenes and i-MXenes, respectively. Interestingly, among all the transition metals used in MXene, Sc, Cr, Y, and W have been reported only in these chemically ordered MXenes [101]. The i-MXenes can be found in two structures: (M′_2/3_M′′_1/3_)_2_C with in-plane elemental order, such as (Mo_2/3_Y_1/3_)_2_C, or M′_4/3_XT_x_ with ordered vacancies, for instance, Mo_1.33_C [29]. Lately, Yang et al. [102] have reported the synthesis of the Mo_1.33_C@R MXene from the in-plane chemically ordered (Mo_2/3_R_1/3_)_2_AlC (*R* = Gd, Tb, Dy, Ho, Er, and Y) MAX phases. The Mo1.33C@Y-HM treated with hydrazine monohydrate demonstrated one of the highest capacitance of about 431 F/g reported so far, which was explained by an increased fraction of the -O groups on the surface. 

Recently, the MXene family has expanded with a new group of members. Before, the most common number of elements in M sites of solid solution MXenes was two. However, in 2021, Nemani et al. [104] demonstrated that it is possible to synthesize high-entropy alloy MAX-phases with respective two-dimensional carbides. They successfully obtained TiVNbMoAlC_3_ and TiVCrMoAlC_3_, which were further used for selective etching of A-layers to preparing TiVNbMoC_3_T_x_ and TiVCrMoC_3_T_x_, respectively. These materials consisted of four different transition metals, forming a high-entropy alloy. MXenes were synthesized using 48 wt.% HF and delaminated with 5 wt.% tetramethylammonium hydroxide. The stoichiometric ratios of M_1_, M_2_, M_3_, and M_4_ transition metals in the synthesized materials were 1:1:1:1 (±0.2). The latter means the formation of high-entropy alloys. They consist of four or more elements with near-equiatomic concentrations of about 5–35 at.%, depending on the number of elements. High-entropy alloys are characterized by forming a single-phase structure and demonstrating unique properties, such as thermal stability and oxidation- and wear-resistance [104,105]. Figure 22 schematically demonstrates the structural features and the process to synthesize high-entropy MAX phases and the respective MXenes. A typical structure for M_4_C_3_T_x_ MXene can be clearly seen in the SEM image (Figure 22e). The XPS of MXenes showed that the in-depth distribution of metal elements was uniform and was found to be about 15 at.%. It was found out that the calculated entropic contribution at 1600 °C had a value of −0.2238 eV/f.u., which was much lower compared to the three-transition metal structure −0.1773 eV/f.u. This finding means that the formation of a single phase is more favorable in four-transition metal MAX than in material consisting of three transition metals. So far, TiVNbMoC_3_T_x_ and TiVCrMoC_3_T_x_ are the first high-entropy MXenes, which were successfully synthesized from MAX phases. The future research is expected to be focused on their properties since such (M_1_M_2_M_3_M_4_)C_3_T_x_ MXenes are expected to possess enhanced characteristics. It means that a new group of high-entropy alloy 2D carbides allows obtaining a vast number of new MXene compositions. They could be suitable candidates for applications in catalysis, harvesting and storage of energy, etc.

### 3.2. Properties

MXene is more attractive than graphene due to its ability to control the properties of the material. This can be done using functionalization, processing, and doping, while graphene allows only functionalization [106].

It was found that the characteristics of MXene can be controlled via the addition of various functional groups, changing the distance between layers, doping in the position of M, A, or X in the corresponding MAX phase, and also adjusting their composition [98]. For example, Anasori et al. showed that surface functional groups have a strong effect on the properties of these two-dimensional materials [11,39]. Using the example of Mo-containing MXene, it was demonstrated that, in the case of the presence of terminal OH and F groups, they behave as semiconductors. In contrast, surface termination with O groups was responsible for the metallic behavior. Doping the MAX phases can make it possible to obtain various solid solutions with new properties that depend on the added elements and their concentrations.

By adjusting the composition of the MAX phase, it is also possible to effectively control the final properties of the material. For example, it was found that the characteristics of Ti_2_AlC can be modified as follows: replacement of Ti and Al with V and Si leads to an increase in Vickers hardness, and the use of N atoms instead of C (Ti_2_AlN) or their combination (Ti_2_AlC_0.5_N_0.5_) allows improving the thermal and electrical properties of the MXenes [107]. 

By doping the MAX phases, solid solutions can be obtained in the M, A, or X positions, making it possible to have properties unique in their functionality. When Sn is added to Layer A, Ti_3_AlC_2_, the oxidation resistance at 900 °C increases by 33% [108]. Due to the possibility of forming solid solutions, an infinite number of non-stoichiometric MXenes can be synthesized [109]. The addition of the necessary transition metals or nitrogen/carbon allows effectively controlling properties of the resulting materials and brings in an attractive possibility of fine-tuning their properties.

The other way to tailor the properties of MXene is to change the inter-flake spacing in the multilayer samples. It can be achieved through intercalation. For instance, adding intercalants affects the MXene conductivity. By nature, MXenes possess metallic conduction, but increasing c-lattice spacing results in semiconductor-like behavior. Hart et al. studied the influence of the intercalation of H_2_O and TBA^+^ on the electric properties of Mo_2_TiC_2_T_x_ [84]. The MXene was annealed in a vacuum to 775 °C in an attempt to investigate the effect of each intercalant. It was found that intercalated Mo_2_TiC_2_T_x_ had a c-lattice parameter of about 37.7 Å and displayed semiconductor-like behavior. After annealing at 200 °C, a de-intercalation of H_2_O was detected, decreasing resistance by 24%. However, at 530 °C, the Mo_2_TiC_2_Tx demonstrated metallic conduction. The resistance dropped by 69%, and the c-lattice spacing was reduced to 24.5 Å. The loss of TBA^+^ intercalant caused such dramatic changes. Therefore, intercalation can be effectively used to modify the properties of MXenes.

Furthermore, the surface functional groups also play an essential role in the performance characteristics of MXenes. In most cases, they are produced by selective etching with HF solution. Therefore, usually, the surface of MXene can be terminated with -O, -F, and -OH-functional species. The latter can affect different properties of MXene, such as electron mobility, 2D ferromagnetism, conductivity, biocompatibility etc. Moreover, recently Kamysbayev et al. presented the covalent functionalization of MXenes through etching in CdCl_2_ molten salt and Lewis acidic CdBr_2_, and synthesized Br-terminated Ti_3_C_2_Br_2_ and Ti_2_CBr_2_ MXenes for the first time [110]. New functional groups can lead to new surface reactions, including exchanging halide ions with other atoms and functional groups. Such a route can allow effectively controlling the structure and properties of MXene. Hart et al. showed that the loss of surface functional groups significantly affects the electronic properties of MXenes [84]. Annealing the Ti_3_CNT_x_ up to 700 °C demonstrated the release of -F species was followed by a simultaneous increase in electronic conductivity. Previous DFT studies have predicted that the functionalization of Ti_3_CNT_x_ with OH, F, and/or O groups dramatically influences the electronic states near the Fermi level (E_F_). According to the density functional theory predictions, the maximum in the density of states (DOS) plot of Ti_3_CN without surface functionalization is located at E_F_. However, in the case of MXene with termination species, the DOS(E_F_) is found to be significantly decreased [111]. Therefore, the change in the electrical conductivity in functionalized MXenes can be related to the decrease in the concentration of carriers. 

### 3.3. Thermal Stability

MXene exhibits limited stability in water and oxygen, and exposure to light can accelerate its oxidation process. Therefore, it is recommended to refrigerate the MXene suspension for storage in an oxygen-free, dark environment. Oxidation begins from the edge of the two-dimensional carbide layer, and the formed nanocrystals of metal oxide TiO_2_ then propagate over the entire surface [92].

In the process of selective etching of atoms of the MXene precursor material, a significant number of carbon atoms are replaced by O^2-^ ions. Oxygen in the crystal lattice of the transition metal carbide acts as a nucleation center for the growth of titanium oxide nanoparticles. When MXene nanolayers come into contact with water, they react with reactive oxygen species (hydroxyl radical or oxygen molecules). This leads to the destruction of Ti–C and the formation of Ti–O bonds, which accelerates the growth of TiO_2_ nanoparticles. The liberated carbon atoms form a disordered carbon material. Prolonged exposure to water or oxygen leads to a gradual growth of titanium oxide nanoparticles, which is accompanied by the destruction of MXene [82].

The most studied representative of two-dimensional carbides, Ti_3_C_2_T_x_, was used as an example to study the stability of MXene in an aqueous colloidal solution at room temperature (23 °C) [112]. Immediately after synthesis, the Ti_3_C_2_T_x_ nanolayers exhibited a clean surface and edges (Figure 23a). This MXene had a hexagonal arrangement of atoms, and their surface was uniform with a transverse dimension of about 600 nm (Figure 23e). After exposure to air at 23 °C for 7 days, some “branching” up to 100 nm was noticeable at the edges of Ti_3_C_2_T_x_, and nanoparticles 2–3 nm in size appeared on the basal planes (Figure 23b,e). The insets to Figure 23e,f show the spatial fast Fourier transform results, which show that the nanoparticles are titanium oxide, TiO_2_. After keeping the colloidal solutions at room temperature for 30 days, complete degradation of MXene was observed. Figure 23c,f shows only large TiO_2_ particles and disordered carbon. The stages of MXene degradation using Ti_3_C_2_T_x_ as an example are schematically shown in Figure 23g.

Next, an interesting experiment was conducted to determine the two parameters that most strongly affect the stability of the MXene; the temperature and the presence of oxygen were chosen as such parameters. The Ti_3_C_2_T_x_ solution was stored in bottles filled with argon to eliminate the influence of oxygen. Compressed Ar prevented air from entering the aquatic environment, keeping the system free from dissolved oxygen. It is also known that lowering the temperature should slow down the rate of the oxidation process [112]. Therefore, Zhang and his colleagues [112] investigated the lifetime of an aqueous solution of MXene also under conditions of reduced temperature (5 °C). It was observed that the samples stored in argon showed superior stability compared to MXene stored in air. Removing dissolved oxygen was more effective at suppressing MXene oxidation than simply lowering the solution temperature. Even after 3 weeks of storage in argon, the degradation of Ti_3_C_2_T_x_ was negligible. However, the combination of argon atmosphere and lower temperatures provided the best storage environment for the layered MXene colloidal solutions. A decrease in the rate of degradation of Ti_3_C_2_T_x_ in argon and at low temperatures suggests that dissolved oxygen is the main oxidant. The aqueous medium has also been found to be a mild MXene oxidizing agent. The reaction of converting Ti_3_C_2_O_2_ into titanium oxide can be written as follows:(15)$${\mathrm{Ti}}_{3}C_{2}O_{2}+4H_{2}O=3{\mathrm{TiO}}_{2}+2C+4H_{2}$$

It is noteworthy that the degradation process of MXene depends on their lateral size, while the stability increases with increasing the thickness of layers. In addition, Ti_3_C_2_T_x_ showed a longer lifetime than Ti_2_CT_x_, which can be explained by an increase in the stability of MXene with an increase in n from 1 to 3 [114].

Zhang et al. [113] demonstrated that it is possible to extend the storage of MXene as an aqueous dispersion without any additives while preventing oxidation from three weeks to 650 days. This was achieved by lowering the temperature of the MXene solution to −20 °C, which can be easily accomplished with conventional freezers in a laboratory setting. Although the color of an aqueous dispersion of MXene depends on the concentration of MXene, a color change from dark gray to white usually indicates an oxidation process [115]. The effect of temperature on the color change of the Ti_3_C_2_T_x_ aqueous solution caused by the gradual degradation of MXene is shown in Figure 24a–c. The color change of the dispersion was clearly visible on the second day at room temperature that continued until the dispersion became utterly white by the 30th day (RT-30) (Figure 24a). Oxidation of MXene is indicated by discoloration, and solution stability was observed to be similar to previous reports of MXene oxidation using different etching conditions.

The effect of high temperatures on the stability of multilayer V_2_CT_X_ was investigated by Wu et al. [116]. The MXene composition remained the same when heated to 375 °C in an argon atmosphere. However, after heat treatment at 1000 °C, V_2_C was transformed into V_2_O_3_ or V_8_C_7_. In the air, the V_2_C MXene structure was only stable up to 150 °C. With an increase in temperature, its surface began to oxidize in air. However, in this case, the oxidation product was V_2_O_5_, not V_2_O_3_, since there was enough oxygen in the air atmosphere. At a temperature of 1000 °C, complete oxidation of the sample was observed, which already consisted of many V_2_O_5_ crystals in the form of rectangular blocks. Oxidation of two-dimensional carbides was also accompanied by the release of -OH/-O/-F functional surface groups and CO_2_. The following equation can represent the oxidation reaction of V_2_CTX MXene:(16)$$V_{2}{\mathrm{CT}}_{x}+3.5O_{2}=V_{2}O_{5}+{\mathrm{CO}}_{2}↑+4T_{x}↑$$

In addition, the structural stability of Mxenes in the temperature range of 500–1000 °C in vacuum was investigated using Ti_3_C_2_T_x_ and Zr_3_C_2_T_x_ [48]. It was found that annealing for 2 hours at 1000 °C resulted in a phase transformation to Ti_3_C_2_. There was a complete transformation of the hexagonal organization into a cubic structure with the formation of TiC. The formation of TiO_2_ nanoparticles was also observed, which could occur due to the presence of O-surface groups or residual oxygen in a vacuum. However, on Zr_3_C_2_T_x_, the sample after high-temperature annealing still retained its integrity, although the removal of intercalated particles and surface groups was observed.

Therefore, oxidation of Mxene can be prevented by using low temperatures, vacuum, or an inert gas atmosphere, and isolating them from water and oxygen. High temperatures cause surface oxidation and the gradual formation of the oxides of the transition metals from which the MXenes consist. Various phase transformations of the structure are also possible. Moreover, the temperature and products of such transformations usually depend on the composition of two-dimensional carbides and atmospheric conditions.

## 4. MXene Applications 

Having received the first representative of the new family of two-dimensional materials with unique properties, MXenes have become one of the most popular areas of modern science [117]. Their active study led to the rapid discovery of new transition metal carbides/nitrides/carbonitrides with various chemical compositions, structures, and characteristics. It has been found that, due to the wide range of properties inherent to MXenes, they can be successfully used for various applications. For example, the outstanding electrical and optical properties of these two-dimensional materials may be of interest to design sensors [118]. They also demonstrated strong absorption in the near-infrared region, low cytotoxicity, high biocompatibility, and even some selectivity towards cancer cells, making them attractive candidates for bioimaging, photothermal therapy, theranostics, and drug delivery in the human body [81]. The development of electrical power systems causes a need for new dielectrics, which must exhibit low dielectric loss and high dielectric constant to be suitable for electrostatic film capacitors with high energy density, flexibility, and high breakdown strength. It was found that incorporation of MXene nanosheets to polymer allows achieving the optimal combination of high dielectric constant and low dielectric loss [119]. For instance, poly(vinylidene fluoride) (PVDF)-based percolative composites with 2D Ti_3_C_2_T*_x_* nanosheets as fillers reached a dielectric constant as high as 10^5^ near the percolation limit (15.3 wt.% MXene) [120]. Moreover, the dielectric loss of the MXene/P(VDF-TrFE-CFE) composite increased from 0.06 to 0.35 (5-fold), but the dielectric constant increased by 25 times within 0–10 wt.% MXene composition range. The origin of the substantial permittivity enhancement is primarily due to the microscopic dipoles formed by the accumulation of charges at the interfaces between the MXene fillers and the polymer matrix. Compared with other fillers (hydrothermally reduced graphene oxide, copper phthalocyanine, functionalized graphene nanosheets), MXenes provided the best dielectric constant/loss factor trade-off. The other example of such composite is multilayered Ti_3_C_2_T_x_/PVDF films fabricated by three steps: spin coating, spray coating, and hot-press methods [119]. This structure consisted of overlapped layers of MXene and PVDF placed on each other to provide enhanced interfacial interactions due to Ti_3_C_2_T_x_ and good charge carrier insulation through the ferroelectric PVDF layer. The dielectric constant of the multilayer 4MXene/5PVDF (four layers of Ti_3_C_2_T_x_ and five layers of PVDF) measured at 1 kHz was about 41, higher than that of pure PVDF (10.5). An increase was caused by enhanced Maxwell–Wagner–Sillars (MWS) interfacial polarization due to the difference in dielectric performances of Ti_3_C_2_T_x_ and PVDF. The dielectric loss at 1 MHz was suppressed below 0.2. Additionally, MXene/PVDF films had low conductivity of <10^9^ S m^−1^ at 1 kHz. The frequency-dependent alternating current (AC) conductivity suggested an insulating behavior of the composite films. It also demonstrated a superior dielectric constant to loss factor ratio of about 1464.3. Therefore, composite MXene-based films are promising broadband dielectric material for high-frequency capacitors. Another example of the use of MXene phases as reinforcement in ceramic composites is Ti_3_C_2_T_x_/Al_2_O_3_, which at 2 wt.% of Ti_3_C_2_T_x_ showed a 300%, ~150%, and ~300% improvement of the fracture toughness, bending strength, and hardness, respectively [121]. The ZnO-Ti_3_C_2_ composite produced by the cold sintering process improved the electrical conductivity of the oxide matrix by 1–2 orders of magnitude and showed a 150% increase in hardness and elastic modulus [122]. The enhancement of mechanical properties of silicon carbide modified with Ti_3_C_2_T_x_ was also reported by Petrus et al. [123]. The Ti_3_C_2_T_x_ MXene composite films with segregated polystyrene inclusions studied by Iqbal et al. [124] showed superior electromagnetic interference efficiency, making them a promising shielding material with tunable electromagnetic wave absorption properties. Furthermore, the combination of MXenes as nanofillers with polymers allows preparing the flame retardant nanocomposites [125]. The unique layered structure and versatile interface chemistry of 2D MXenes make it possible to improve polymer properties [126]. For instance, the addition of 2.0 wt.% Ti_3_C_2_Tx into unsaturated polyester resin (UPR) resulted in the formation of nanocomposite with an enhanced fire safety property: the peak heat release rate, the total smoke production, and carbon monoxide production were reduced by 29.56%, 25.26%, and 31.58%, respectively [127]. Yu et al. manipulated the surface of Ti_3_C_2_T_x_ with 3-aminopropylheptaisobutyl-polyhedral oligomeric silsesquioxane (AP-POSS) through electrostatic interactions and incorporated the POSS-Ti_3_C_2_T_x_ into polystyrene (PS). The resultunt PS nanocomposites exhibited 39.1%, 54.4%, and 35.6% reductions in the peak heat release rate, the CO production rate, and the CO_2_ production rate, respectively [128]. Thus, MXenes are popular materials with many possible applications in various industries. Some of them are schematically shown in Figure 25.

### 4.1. Biomedicine

Two-dimensional MXenes are being actively investigated for their potential medical applications. For example, the large surface area of their ultrafine planar structure provides numerous anchorage sites for drug molecules. MXene nanosheets inherently possess excellent photothermal converting properties that can be further developed as photothermal nanoagents for cancer hyperthermia when exposed to external near-infrared (NIR) laser irradiation. In recent years, MXenes have been applied as agents for photo-thermal therapy (PTT), as a bioimaging technology, for drug delivery, and as an antibacterial substance and electroconductive base in biomaterial applications. In the current review, we cover two of the most popular applications of MXenes: a cancer treatment strategy and an antibacterial/antiviral therapy.

#### 4.1.1. Photo-Thermal Therapy

Photo-thermal therapy (PTT) emerge as promising non-invasive and light-triggered anti-cancer approaches [129]. PTT generates heat by transforming near-infrared (NIR) light into thermal energy in a minimally invasive manner, which is regarded as the potential treatment substitution for conventional clinical cancer therapeutic modalities [130]. Gold nanoparticles, graphene, transition metal dichalcogenides, black phosphorus, and some organic nanosystems are promising agents for PTT [131]. MXenes are the wide-spectrum photothermal-conversion agents that can be activated in both NIR-I (750–1000 nm) and NIR-II (1000–1350 nm) biowindows that allow maximal penetration to the living tissues [132]. NIR-II demonstrates higher maximum permissible exposure and penetration depth than the NIR-I [133], but their application depends on the extinction coefficient on MXenes.

Ti_3_C_2_ was the first MXene applied for PTT both in vitro and in vivo [134] due to its high extinction coefficient and low toxicity. Ti_3_C_2_ exhibits a high extinction coefficient of 25.2 L g^−1^ cm^−1^, which is remarkably higher than that of traditional Au nanorods (13.9 L g^−1^ cm^−1^) [135], indicating a favorable NIR laser-absorption property of Ti_3_C_2_ nanosheets. Lin at all. first described the possible application of Ti_3_C_2_ for in vitro and in vivo photothermal ablation. They demonstrated high biocompatibility of a Ti_3_C_2_–soybean phospholipid (SP) complex in concentration up to 400 μg/mL using 4T1 cancer cell line. 4T41 cells were incubated with Ti_3_C_2_–SP nanosheets at varying concentrations (0, 6, 12, 25, 50, and 100 μg/mL) for 4 h and then exposed to the 808 nm laser at a power density of 1.0 W cm^−2^. They proved the photothermal ablation effect of Ti_3_C_2_–SP that strongly depends on MXene concentration [134]. The in vivo investigation demonstrated 1.73% Ti_3_C_2_–SP accumulation in tumors within 24 h after intravenous injection on nude mice with no adverse effect on living organs. The tumor on the Ti_3_C_2_–SP–NIR group was completely eliminated after 16 days of observation. In 2017, Liu et al. established Ti_3_C_2_ multifunctional nanoplatforms via layer-by-layer surface modification with doxorubicin (DOX) and hyaluronic acid (HA) [136]. The Ti_3_C_2_–DOX complex demonstrates potential application for combine the PTT-chemotherapy approach with high biocompatibility, tumor-specific accumulation, and stimuli-responsive drug release behavior. They demonstrate no toxicity for both unmodified Ti_3_C_2_ nanosheets and Ti_3_C_2_–DOX complex in concentration up to 200 μg/mL. 808 nm laser irradiation (0.8 W cm^−2^) demonstrated HCT-116 cells death both from PTT effect and DOX release because of the disruption of electrostatic interaction in the Ti_3_C_2_–DOX system. Xiaoxia Han in 2018 reported a similar PTT effect of DOX–Ti_3_C_2_–SP complexes against 4T1 timer cells in vitro as well as high effectiveness on 4T1 tumor-bearing nude mice [137]. A NIR 808 nm laser with 1.5 W cm^−2^ power after 10 min irradiation leads to tumor reduction in the DOX–Ti_3_C_2_–SP group. The authors demonstrated that Ti_3_C_2_ MXene had a high drug-loading capability (211.8%) and exhibited both pH-responsive and NIR laser-triggered on-demand drug release. Later, some authors showed different Ti_3_C_2_ modification and high PTT effectiveness in NIR-I region of MXene-cellulose-DOX complexes [138], Plasmonic MXene-based nanocomposites (Au/MXene and Au/Fe_3_O_4_/MXene) [139], Ti_3_C_2_–cobalt nanowire–DOX construction [140], Ti_3_C_2_–metforminpolysaccharide composites [141] and Ti_3_C_2_–superparamagnetic iron oxide nanoparticles [142]. All these researches demonstrated the possible applications of Ti_3_C_2_ MXenes for PTT and as a promising platform for combined photo-dynamic and chemotherapy. 

In contrast with Ti_3_C_2_, Nb_2_C MXenes exhibit effectiveness in both NIR-I and NIR-II regions. In 2017, Lin et al. described a highly efficient Nb_2_C–PVP nanosheet in vivo photothermal ablation of mouse tumor xenografts in both NIR-I and NIR-II windows. They used 4T1 cells as cancer model and 5 min of 808 nm and 1064 nm laser irradiation after MXene-cell co-incubation. Nb_2_C-PVP intravenous injection on the 4T1 tumor-bearing mice with next NIR-I and NIR-II irradiation within 10 min led to significant tumor size reduction compared to control groups [143]. A year later, Han et al. demonstrated the synthesis of Nb_2_C–arginine–glycine–aspartic pentapeptide c(RGDyC) and its effectiveness in NIR-II PTT against U87 cancer cell line and corresponding tumor xenograft with 92.7% inhibition efficiency [132]. In 2019, Xiang and co-authors engineered a free radical nanogenerator via direct growth of mesoporous silica layer onto the surface of two-dimensional Nb_2_C MXene nanosheets toward multifunctionality, where the mesopore provided the reservoirs for initiators, and the MXene core acted as the photonic thermal trigger at the NIR-II region. The 1064 nm laser irradiation induced the fast release and quick decomposition of the encapsulated initiators (AIPH) to produce free radicals that provided additional effects to 4T1 cancer cells [144].

Some research demonstrated the possible application of V_2_C and Ta_4_C_3_ MXene for PTT cancer treatment. Zada et al. demonstrated high photothermal conversion efficiency of V_2_C MXenes with effectiveness against MCF-7 cancer cell line and MCF-7 tumor-bearing mice in the NIR-I region (808 nm). Additionally, they proved that significant photoacoustic (PA) and magnetic resonance (MR) signals were observed even at low concentrations in vitro and in vivo that open perspectives for bio-imaging [145]. Zhuang Liu in 2018 demonstrated the possible application of Ta_4_C_3_ MXenes for theranostics. Functionalized superparamagnetic iron-oxide MXene composite (Ta_4_C_3_-IONP-SPs) exhibited high performance for contrast-enhanced CT imaging and significant effectiveness against 4T1 breast cancer cells after irradiation in NIR-I (808 nm) regions.

Based on available data, there are two possible mechanisms of direct MXene anticancer PTT. Cell membrane adsorption of MXenes and their penetration inside the cancer cell lead to direct cell death after photothermal conversion of NIR light (Figure 26a). The heating of intercellular media leads to mediated cell death via the initiation of apoptosis (Figure 26b). Chemotherapeutic drugs and nanoparticles loaded to MXene can provide additional effects and secure complete anticancer therapy. 

To summarize, a variety of MXenes demonstrated the ability for PTT as is and as a part of complexes, including chemotherapy agents, bioactive substances, and magnetic nanoparticles. Most research proved low toxicity and high effectiveness against different cancer cell lines both during in vitro and in vivo experiments. NIR-I and NIR-II laser irradiation could be applied for PTT with respect to MXene nature, but most research used an 808 nm wavelength. Considering the high loading capacity of MXene and the active surface, chemotherapeutic and ROS agents could be used for combined chemotherapy and PDD approaches. Loading MXenes with specific anti-cancer antibodies can provide a new approach in target cancer PTT with minimal side effects and MXene accumulation in tissues. Despite a large amount of research, the biological effect of MXenes is still under question, including full metabolic fate, long-term side effects, and influence on immune cells. 

#### 4.1.2. Antibacterial Activity

The first report about the antibacterial activity of MXene was presented by Y. Gogotsi and co-authors in 2016 [146]. They utilized *Escherichia coli* (E. coli) and *Bacillus subtilis* (B. subtilis) strains to assess the antibacterial activity of Ti_3_C_2_T_x_ MXenes. 100 µg/mL Ti_3_C_2_T_x_ within 4 h of exposure demonstrated more than 98% effectiveness against both Gram-negative and Gram-positive bacteria. Authors clearly observed dose-depending bacteria effectiveness for both strains with more substantial influence to Gram-positive ones that could relate with more negatively charged surface of Gr(+) bacteria wall. Using SEM, the authors showed severe membrane disruption and cytoplasm leakage after bacteria co-cultivation with 100 µg/mL of Ti_3_C_2_T_x_. HRTEM demonstrated the presence of the highly crystalline Ti_3_C_2_T_x_ layers that could directly affect the cell wall. Additionally, the authors hypothesized that, due to the high conductivity of MXenes, a conductive bridge over the insulating lipid bilayer could form that mediated electron transfer from bacterial intracellular components to the external environment and caused the bacteria cell death. Later, the same group developed a micrometer-thick titanium carbide (Ti_3_C_2_T_x_) MXene membranes prepared by filtration on polyvinylidene fluoride (PVDF) support for applications in water/wastewater treatment. The membrane provided 73% effectiveness against B. subtilis and 67% against E. coli [147]. It is remarkable that the aged Ti_3_C_2_T_x_ membrane increased effectiveness up to 99% against both microbial strains.

Mayerberger et al. produced an electrospun membrane with encapsulated delaminated Ti_3_C_2_T_z_ (MXene) flakes within chitosan nanofibers for passive antibacterial wound dressing applications. 0.75 wt.% Ti_3_C_2_T_z_- loaded nanofibers provided 95% effectiveness against Escherichia coli and Staphylococcus aureus with no cell toxicity [148]. 

Jastrzębska et al. studied the antibacterial activity of Ti_2_C and Ti_3_C_2_ against E. Coli using the classical culture method. They did not find any bacteria inhibition with Ti_2_C flakes and only a small inhibition zone (up to 1.5 mm) around the Ti_3_C_2_ MXenes [149]. From our experience, the contact inhibition after direct MXene-cell membrane interaction that was proved in [150] should be noted. Later, using a broth microdilution assay, Shamsabadi et al. proved that direct contact between MXene and cell membrane plays a crucial role in the antibacterial mechanism of action. They demonstrated that the sharp age of small size MXene (lateral size less than 0.57 μm) leads to bacterial cell wall damage and DNA release from the cytosol [151]. Another group stabilized Ti_3_C_2_ MXene surface using poly-L-lysine (PLL) that changed the negatively charged cell surface to a positive value and stabilized colloidal solution of MXenes. These changes provided an additional antibacterial effect of MXene solution towards E. Coli compared to non-modified Ti_3_C_2_ [152]. Another approach used to enhance MXene antibacterial properties was implemented by Zheng et al. through conjugating ultra-small gold nanoclusters (AuNCs) on Ti_3_C_2_ nanosheets. They proved an synergic antimicrobial mechanism via mechanical damage of the cell membrane by MXenes and oxidizing of bacterial membrane lipid and bacterial DNA for violent fragmentation via reactive oxygen species generated by AuNCs [153]. Pandey et al. proposed MXene modification with Ag nanoparticles (Ag@MXene) as an alternative for water purification membranes. Their 21% Ag@MXene composite membrane exhibited 99% effectiveness against the E. Coli strain compared to the 60% effectiveness of a pure MXene-loaded membrane. This synergetic effect should be used for water purification and biomedical applications [154].

Another strategy for MXene exploitation as antibacterial agents is applying NIR light to provide photo-thermal ablation. Wu et al. demonstrated high effectiveness for Ti_3_C_2_ MXene combined with 808 nm light (20 min irradiation) against 15 bacterial spices [155]. Remarkable that this method demonstrates effectiveness against methicillin-resistant *Staphylococcus aureus* (MRSA) and vancomycin-resistant *Enterococci* (VRE) as well as against MRSA biofilms.

The latest research demonstrates the antiviral and immunomodulatory potential of MXenes [156]. Unal et al. tested four MXenes, Ti_3_C_2_T_x_, Ta_4_C_3_T*_x_*, Mo_2_Ti_2_C_3_T*_x_*, and Nb_4_C_3_T*_x_*, as antiviral agents against four viral clades. They found that Ti_3_C_2_T_x_ reduced infection in SARS-CoV-2/clade GR-infected Vero E6 cells. Mo_2_Ti_2_C_3_T_x_ also exhibited antiviral activity compared to the Ta_4_C_3_T_x_ and Nb_4_C_3_T_x_ ones. Additionally, they tested MXene interaction with 17 subpopulations of human primary immune cells and concluded that all tested MXenes had excellent bio and immune compatibility with the ability to reduce the release of pro-inflammatory cytokines. 

Based on the abovementioned data, we summarized the possible antibacterial mechanisms of MXenes in Figure 27. Nevertheless, in conclusion, it should be noted that antibacterial activity was proved only for Ti_3_C_2_ MXenes against a few bacterial spices. A more detailed study of the antibacterial spectrum is needed to ensure precise bactericidal mechanisms. Detailed investigation of toxicity/bactericidal balance must also be performed using different MXene compositions. The optimal parameters, including size, composition, and additives, need to be clarified to open clinical application.

### 4.2. Ecological/Environmental Applications

With the rapid growth of the world population and the rise of agriculture and industrial production, a shortage of drinking water is predicted [157]. Therefore, the development of new methods for obtaining clean, fresh water is one of humankind’s most important global problems [158]. To improve water quality, it is necessary to remove harmful impurities that can negatively affect human health and the environment. For example, some of the most common pollutants are heavy metal ions, toxic gases, dyes, hormones, and organic solvents [157]. Among all water pollutants, heavy metal ions (Pb (II), Cr (VI), Hg (II), and U (VI)) are in first place in terms of toxicity to the human body. It is also important to purify water from various organic dyes that are difficult to biodegrade due to their stable molecular structure [159]. Recently, photothermal conversion, membrane separation, and adsorption have become the most popular water purification methods [160,161,162].

#### 4.2.1. Photothermal Conversion

Studies show that natural renewable energy sources such as solar radiation can be effectively used for the purification, distillation, and desalination of seawater [163]. For such purposes, solar steam generators based on the principle of photothermal conversion have demonstrated their effectiveness. In the classical solar steam generators, solar energy is absorbed by the receiver and is then used to heat up the main fluid due to thermal conductivity and thermal convection [157,164]. The coefficient of performance (COP) of such systems is less than 50% since most of the absorbed solar thermal energy is used to heat the main fluid. For efficient water evaporation due to light absorption and reduction of heat losses in the system, the evaporator must have high hydrophilicity and photothermal conversion, excellent light absorption characteristics, and a porous structure to ensure sufficient water supply to the absorbers [165]. In addition, photothermal materials should not sink the water since placement directly at the water–air interface reduces heat loss to bulk water. Recent studies have shown that MXene exhibits high efficiency in absorbing solar energy and has great potential for solar steam generator applications [166].

For example, in 2017, the most popular representative of the MXene family, Ti_3_C_2_, was studied as a solar photothermal material [166]. For its synthesis, the Ti_3_AlC_2_ MAX phase was used, which was treated with an aqueous solution of HF for selective etching of aluminum, followed by immersion in DMSO and sonication. After that, the aqueous suspension of MXenes was filtered through a hydrophilic membrane of polyvinylidene fluoride (PVDF) 0.22 μm thick. Thus, a thin layer of stacked MXene sheets was formed on the surface of the PVDF substrate. To test the characteristics of photothermal water evaporation, the MXene–PVDF photothermal membrane was modified by adding functional groups using a solution of polydimethylsiloxane (PDMS) in hexane. Ti_3_C_2_ demonstrated close to 100% light to heat conversion and excellent light absorption. In this case, the wavelength of the laser source, 473 or 785 nm, did not adversely affect the efficiency of titanium carbide in any way, which indicates the ideal characteristics of photothermal conversion. In addition, a thin MXene-PVDF membrane with a thermal barrier provided a photothermal evaporation efficiency of about 84%. The disadvantage of such a membrane was that it was wetted in water, although, after modification with PDMS, it could float on the surface of water on its own. The non-wetting surface naturally repels water and prevents it from entering the membrane during desalination with the help of solar radiation [167]. Such high characteristics of Ti_3_C_2_ suggest that MXene-based membranes can provide effective water purification from contamination and environmentally friendly long-term operation [168].

Later, in 2018, a self-floating solar steam generator for water desalination based on a hydrophobic membrane with Ti_3_C_2_ was presented [167]. Two-dimensional titanium carbide nanosheets were synthesized by etching the Ti_3_AlC_2_ MAX phase with an HCl/LiF solution. The MXene membrane was prepared by filtering delaminated Ti_3_C_2_ nanosheets modified with a hydrolysis solution of perfluorodecyltrimethoxysilane (PFDTMS) (0.5 vol.% acetic acid, 2.0 vol.% PFDTMS, and 97.5 vol.% isopropanol) onto a mixed ester membrane cellulose with a pore size of 0.22 μm. The solar steam generator device consisted of three components: membrane with Ti_3_C_2_ on filter membrane (solar energy absorber, vapor generator, and salt blocker), commercial polystyrene foam (heat insulator and flotation device), and nonwoven wicks (waterway due to capillary effect). To test the effectiveness of the hydrophobic MXene membrane, a hydrophilic counterpart was also prepared. Modification of Ti_3_C_2_ with PFDTMS led to the formation of functional groups on the -CF_3_ surface, which were responsible for the hydrophobic character. Figure 28 shows a schematic illustration of the evaporation process of hydrophilic and hydrophobic Ti_3_C_2_ membranes. In the case of a hydrophilic membrane, the capillary effect facilitates seawater penetration, leading to the continuous crystallization of salts on the membrane surface with water evaporation. The accumulation of large amounts of salt can cause severe damage to the membrane, as shown in Figure 28a. However, a hydrophobic membrane is non-wettable due to its ability to block water together with dissolved salts under the membrane while vapors freely leave its pores (Figure 28d). Therefore, using a hydrophobic Ti_3_C_2_ membrane in a steam generator device promotes long-term and stable desalination using solar energy. For example, the concentration of the four primary ions (Na^+^, K^+^, Mg^2+^, and Ca^2+^) can be markedly reduced to the level of 99.5%, which indicates effective desalination of seawater. It was also found that the membrane was able to efficiently vaporize typical contaminants (filtration rate of about 100%) that may be contained in wastewaters, such as organic dyes, reactive dark blue (RhB) and methyl orange (MO), heavy metal ions, Cu^2+^ and Cr^6+^, and other organic substances, including benzene and acetone (Figure 28f).

Ju et al. [162] presented a macroporous three-dimensional MXene architecture (3DMA) for high-efficiency solar steam production. Using a two-step immersion method, such structures were obtained by applying Ti_3_C_2_T_x_ MXene layers onto a melamine foam (MF) scaffold. At the first stage, the MP was immersed in polyvinyl alcohol (PVA) solution and dried in a vacuum. At the second stage, the resulting MA/PVA composite scaffold was immersed in a suspension of MXenes, followed by vacuum drying. PVA acted as an adhesion promoter between MXene and the scaffold. Figure 29a,b shows the difference in the morphology of melamine foam before and after modification of Ti_3_C_2_T_x_. A decrease in pore size was observed as the MXene nanolayers were collected on the scaffold walls in the form of large chunks. To assess the efficiency of solar steam generation, the resulting porous structure was placed at the water–air interface while it floated freely on the water surface without any support. The efficiency of 3DMA for a solar radiation intensity of 1000 W·m^−2^ was 82.4% at an evaporation rate of 1.309 kg·m^−2^·h^−1^, and, at an illumination of 5000 W·m^−2^, it reached 88.1% at an evaporation rate of 6.997 kg·M^−2^·h^−1^ [165]. The resulting three-dimensional structure was effective for the desalination of seawater, which was manifested by a significant decrease in the concentration of Na^+^, K^+^, Mg^2+^, and Ca^2+^ cations (Figure 29c). In addition, when treating wastewater from methyl orange and methylene blue (MB) dyes, 3DMA demonstrated a capture rate close to 100%, as shown in Figure 29d.

Additionally, a steam generator based on nanocoatings of two-dimensional Ti_3_C_2_T_x_ MXene was described for the first time, which imitated the hierarchical textures of black scales of the West African gaboon viper (*Bitis rhinoceros*) [169]. MXene nanolayers were obtained by etching the Ti_3_Al_2_C_2_ MAX phase in a LiF-HCl fluoride solution followed by ultrasonic-assisted delamination. The aqueous MXene solution was then sprayed onto thermosensitive polystyrene substrates and air-dried. Then, the samples were warmed up to a temperature above the glass transition temperature of polystyrene while still below 140 °C to exclude degradation of MXene. This caused them to shrink to ≈ 50% of the original length and ≈ 25% of the original area. Repeating the described deformation process several times, a biomimetic hierarchical structure of MXene was obtained, where the primary crumpled structure mimicked the micro-ridges of the black scales of vipers and the secondary structureand the secondary structure (more minor folds) resembled the texture of nano-ridges. The presence of such a branched texture of biomimetic Ti_3_C_2_T_x_ nanocoatings promoted multiple scattering and reflection of light, causing broadband light absorption (84.9–86.9%) and improved equilibrium temperature at illumination of 1000 W·m^−2^ (58.2–62.6 °C) compared to conventional flat samples MXene (46.8–64.0% and 50.4–58.1 °C). An increase in the complexity of the structure (degree of deformation) of the hierarchical nanocoating caused a further increase in light absorption and thermal conductivity up to 93.2% and 65.4 °C, respectively. Solar steam generating devices based on such biomimetic MXene nanocoatings demonstrated a high evaporation rate of about 1.33 kg·m^−2^·h^−1^ at a solar radiation intensity of 1000 W·m^−2^. It has also been shown that they can be used to implement flexible solar/electric heaters. Consequently, the MXene surface treatment technology had a significant impact on the final performance of the device.

In 2020, Zhao et al. reported a new strategy for highly efficient solar energy conversion to steam using composite polydopamine@Ti_3_C_2_T_x_ (PDA@MXene) microspheres [170]. Mixing a colloidal solution of exfoliated MXene and an aqueous dispersion of PDA led to the formation of many aggregates, which were self-assembled PDA microspheres wrapped in Ti_3_C_2_T_x_ (PDA@MXene). Numerous hydrogen bonds ensured the adhesion between nanosheets and microspheres. The photothermal layer of composite microspheres demonstrated a synergistic effect in efficiently absorbing solar radiation and converting solar radiation into steam. The steam generation efficiency at an illumination of 1000 and 4000 W·m^−2^ reached 85.2% and 93.6%, respectively. In addition, the PDA@MXene photothermal layer allowed the production of pure water from seawater with a salt capture rate of over 99%. The maximum rate of water evaporation using a PDA@MXene membrane with a loading mass of microspheres of about 0.8 mg/cm^2^ was 1.276 kg·m^−2^·h^−1^ at a radiation intensity of 1000 W·m^−2^. The photothermal layer of the composite was mechanically stable and effectively maintained stable light absorption and water evaporation characteristics after several lighting cycles. However, experiments with seawater desalination showed that the surface of the hydrophilic membrane becomes contaminated after 10 h of desalination due to salt crystallization. In this case, washing off these crystals did not affect the subsequent characteristics of the evaporation of membranes based on composite PDA@MXene microspheres.

Thus, it can be concluded that MXenes, with their unique and customizable properties, provide new strategies for efficient water purification and desalination. So far, only Ti_3_C_2_T_x_ was used as a photothermal material. However, strong experimental results suggest that other members of this large family of two-dimensional materials will also be effective and possibly exhibit improved light absorption and water evaporation characteristics. Therefore, MXenes show tremendous potential in generating solar power in steam to solve the problem of drinking water shortages.

#### 4.2.2. Adsorption

In addition to other strategies, MXenes are actively used for another popular water purification method, namely adsorption. This is due to its simplicity, economy, and the absence of secondary pollution [171]. Adsorption is considered the most effective method for removing heavy metal ions because other strategies such as biological processing or chemical reactions cannot metabolize them or break them down. However, for the material used as an adsorbent to be effective, it has to have a large surface area and high functionality [172].

It was found that titanium exhibits a high sorption affinity for metal ions [173]. Therefore, in 2014, Peng and colleagues [135] investigated the sorption capacity of Ti-based MXene to purify water from lead ions. They suggested that, since the surface of individual Ti_3_C_2_ nanolayers possesses terminals with functional OH or F groups, the substitution of the cation in the hydroxyl group can lead to the activation of ion exchange sites, providing sites for sorption of toxic metals. Moreover, the large area of the layered MXenes promotes the binding of metal ions that have to be removed [174]. For the sorption of Pb (II), the authors proposed Ti_3_C_2_(OH/ONa)_x_F_2−x_, which was synthesized by selective etching of the Ti_3_AlC_2_ MAX phase followed by alkalinization–intercalation. As a result of the treatment, some of the terminal surface F groups were partially replaced by OH groups, and OH groups were converted to ONa groups. The results of kinetic tests showed that sorption equilibrium was reached in just 120 s. In addition, Ti_3_C_2_(OH/ONa)_x_F_2−x_ made it possible to lower the Pb (II) content in wastewater to levels below 10 μg/L. Moreover, the adsorption properties of the material were reversible. Such a high sorption capacity was associated with hydroxyl groups in activated Ti sites, where the Pb (II) ion-exchange was facilitated by forming a hexagonal potential trap. However, the residual amount of F groups weakened the adsorption of lead cations [175].

Later, papers were published in which the use of MXenes for water purification by adsorption from other toxic metals was investigated. For example, Ti_3_C_2_T_x_ nanolayers synthesized by selective etching of Ti_3_AlC_2_ in a 10% HF solution followed by ultrasonic delamination have demonstrated excellent ability to remove Cr (VI) [176]. Hexavalent chromium is a strong oxidizing agent that is highly toxic to plants, animals, and humans. Improper disposal of wastewater after various industrial processes causes severe groundwater pollution and damages the environment [176]. The maximum removal capacity for Cr (VI) was 250 mg/g. At the same time, the residual concentration of hexavalent chromium after purification (less than five parts per billion) was much lower than the standard for drinking water recommended by the World Health Organization (0.05 parts per million). Removal of Cr (VI) by two-dimensional Ti_3_C_2_T_x_ nanolayers occurs via a reduction reaction. MXenes reduce Cr (VI) to a less toxic form of Cr (III), which then can be removed without alkaline treatment at an optimum pH of 5.0. It has also been shown that Ti_3_C_2_T_x_ can be used for the reductive removal of other oxidants such as K_3_[Fe(CN)_6_], KMnO_4_, and NaAuCl_4_.

In 2016, Wang and co-authors demonstrated for the first time the use of multilayer V_2_CT_x_ MXene as a potentially effective adsorbent for U (VI) [177]. Efficient treatment of nuclear waste is becoming a serious environmental problem, as long-lived actinide contamination threatens the environment even in small quantities due to their radiological and chemical toxicity. Therefore, the use of MXene as an adsorbent for trapping radionuclides is a promising direction. The V_2_CTx powder was obtained by immersing the V_2_AlC MAX phase powder in a 40% concentrated HF solution, followed by stirring. V_2_CT_x_ was generally stable in the process of capturing U (VI) from aqueous solution in the pH range from 3.0 to 5.0, but its sorption behavior showed a significant dependence on pH. The best results were obtained at pH 5.0. For example, the maximum absorbent capacity of a powder containing 32 at.% of the unreacted V_2_AlC powder was 174 mg/g. The adsorption capacity of pure two-dimensional MXene nanolayers reached about 256 mg/g. The weak binding of uranium ions at acidic pH levels (lower than 3) also suggests that multilayer V_2_CT_x_ can be regenerated with acidic solutions. Due to functional -OH/-O and -F surface groups, which play a role in the heterogeneous adsorption sites, the capture of U (VI) can be described by a heterogeneous adsorption model. The results of experiments and theoretical calculations showed that V_2_CT_x_ adsorbs uranium ions by forming bidentate adsorption configurations with hydroxyl groups attached to V atoms. The adsorption process proceeded according to the ion exchange mechanism, which was confirmed by deprotonation of the hydroxyl group after binding to U (VI).

In addition to Pb (II), Cr (VI), and U (VI), Fu et al. showed in 2020 that MXene could also adsorb Hg (II) [178]. Since mercury, which is one of the most toxic heavy metals, is widely used in the process of extraction of precious metals, e.g., from electronic devices and industrial catalysis, excessive environmental pollution causes irreparable harm to human health and natural ecosystems [179,180]. For practical use, it is desirable that the adsorbent can effectively remove Hg (II) ions from water even under harsh chemical conditions. The acid–base resistance of M-Ti_3_C_2_ nanosheets was evaluated at various pH values ranging from 1 to 12. M-Ti_3_C_2_ nanosheets (dosage 0.1 g/L) were able to remove less than 10% Hg (II) at pH ≤ 2.0. In contrast, the removal efficiency increased significantly at higher pH, and more than 99.0% of Hg (II) was removed over a wide pH range from 4 to 10. Moreover, M-Ti_3_C_2_ showed high removal efficiency (>97.0%) even in highly alkaline environments (pH > 11).

In parallel with the first experiment to remove Pb (II), it was demonstrated in 2014 that Ti_3_C_2_T_x_ could also be effectively used for the adsorption of the cationic dye methylene blue (MB) [181]. The reaction of the MXene with an aqueous solution of MB under ambient conditions was described by three stages: (1) active adsorption of MB molecules on the surface of Ti_3_C_2_T_x_, (2) an increase in the disordered packing of layers, and (3) oxidation of the MXene.

It has also been demonstrated that multi-layer V_2_CT_x_ MXene could be successfully used as an adsorbent for removing colorants in wastewaters. Usually, dyes in water are found in the form of ions, so they can be removed using special adsorbents. More efficient adsorption of V_2_CT_x_, compared to the most studied Ti_3_C_2_T_x_ could be explained by its simpler unit cell and greater distance between the layers [182].

Due to their unique intrinsic properties, MXenes are of greater interest as adsorbents than graphene oxide, which consists of only one element and has a simpler and insufficiently functionalized surface and a high cost [175,183]. A large surface area, an abundance of active sites, and good dispersibility and reducibility, as well as the fact that many cations could spontaneously intercalate between layers of two-dimensional MXenes, make them promising candidates for water purification from toxic oxidants associated with heavy metal ions, radionuclides, and organic dyes [184]. To date, only two representatives of a large family of these materials, Ti_3_C_2_T_x_ and V_2_CT_x_, have been studied as adsorbents. However, the emergence of new work with other MXenes is expected to substantially enrich the fundamental understanding of adsorption mechanisms and significantly expand the range of their possible applications for environmental cleaning.

### 4.3. Multifunctional MXene-Based Smart Textiles

The range of MXene applications is extensive, and it is hard to describe all of them in one article. Therefore, we would like to briefly mention one more interesting field where these 2D materials can be effectively implemented: wearable electronics with multifunctional characteristics. Lately, it has been shown that wearable and flexible textile-based electronics can be extremely useful due to their versatility. Such smart textile devices can be successfully applied for healthcare electronics [185], human motion monitoring [186], antibacterial [187,188], pressure sensors [189], electromagnetic interference (EMI) shielding [190], and so forth. Although multifunctional textiles are excellent candidates for wearable electronics, they have some limitations. They are connected with difficulties integrating necessary functions, such as conductivity, into the traditional textile substrates while maintaining their high intrinsic properties: breathability and flexibility [191,192]. Therefore, a search for new conductive materials that can be used for smart textiles has led to the recently discovered two-dimensional MXenes. Their unique properties allowed fabricating electronic textiles demonstrating multifunctionality. 

For instance, Zhang et al. [192] reported preparing the MXene-decorated woven cotton fabrics with an interwoven conductive network. Such textile maintained the flexibility and air permeability of cotton and, in addition, gained new exciting characteristics even at the low addition of Ti_3_C_2_T_x_ nanosheets. The synthesized MXene-decorated fabric showed remarkable EMI shielding, Joule heating, and strain sensing performances. The cotton textile with vertically interwoven weft and warp yarns was used. That allowed obtaining the vertically interconnected conductive networks that could increase the MXene conductivity. Meanwhile, the Ti_3_C_2_T_x_ nanosheets were produced by selective etching of Ti_3_AlC_2_ MAX-phase in the HCl + LiF solution. Synthesized MXenes were then uniformly deposited onto the cotton fabric with 2−6 wt.% concentration by the spray-drying coating method. The study showed that such modified fabrics could be successfully applied as flexible heaters for warming up any bendable parts of the human body due to their outstanding Joule heating performance with a temperature up to 150 °C at an external voltage of 6 V. The MXene loading content and voltage could easily tailor the temperature of such heaters. Moreover, MXene-decorated fabric was found to be a promising candidate for the application in sensing small-range human activities since sensors on its bases exhibited high stability after 5000 cycles and superior sensitivity even at a bending strain of about 0−2.09%. MXenes with high electrical conductivity of about 48−5 Ω sq^−1^ made it possible to turn simple cotton fabrics into multifunctional textiles, which can be used for wearable electronics.

Another way to employ MXene-based flexible material for electronic device-cooling applications was proposed by Liu et al. [193]. The authors produced a free-standing graphene/MXene film, demonstrating excellent thermal conductivity and heat dissipation for fire retardant ability. This composite film was produced from graphene oxide (GO), synthesized by modified Hummer’s method [194], and Ti_3_C_2_T_x_, prepared by selective etching of T_i3_AlC_2_ in HCl + LiF mixture. The vacuum-assisted filtration with a cellulose ester filter membrane was used to synthesize graphen oxide/Ti_3_C_2_T_x_ film. After that, to weld the superior flexible composite graphene/MXene (GM) paper, it was reduced through immersion in hydroiodic acid, placed in a warm oil bath at 90 °C, dried at 40 °C, heated to 473 and 1000 K in an Ar atmosphere, and finally cooled down. The GM films demonstrated high thermal conductivity. At the concentration of MXene of about 40 wt.%, the thermal conductivity reached up to 26.49 W m^−1^ K^−1^, which was 6.71-fold and 2.93-fold higher than graphene oxide and reduced graphen oxide films. After welding graphene-MXene, more effective formation of paths for heat conduction in graphene was observed. Thus, a few reasons contributed to enhanced thermal conductivity of composites: the sp2 carbon structure of graphene and orderly alignment of MXene in the graphene matrix. Moreover, the experiment when GM paper with 40 wt.% of MXene was used as ultra-thin heat sink for LED was conducted. The results showed that a protective layer of nonconductive TiO_2_ and graphite, appeared during combustion, provided a good flame retardant ability and outstanding heat-dissipation.

Apart from spray-drying [190,191] and vacuum-assisted filtration [195] strategies, there are also dip-coating [196] and 3D printing [197] methods. An example of the dip-coating approach is a superhydrophobic and breathable intelligent textile device with a four-core shell structure [196]. The consisting parts are the following: (1) a cheap modified elastic polypropylene textile substrate, (2) an interface polydopamine (P) layer for creating active sites for the MXene and better bonding between the MXene layer and substrate, (3) a functional MXene layer (M) of Ti_3_C_2_T_x_ nanosheets, and (4) a protective polydimethylsiloxane (PDMS) layer for decreasing the MXene surface energy, preventing their oxidation, and providing a superhydrophobic ability. All the steps were performed by the dip-coating method through immersion to a certain solution regarding the layer. Figure 30 demonstrates possible applications of a fabricated MXene-based PM/PDMS textile. This composite is expected to remarkably perform as waterproof all-in-one wearable electronics capable of tracking body motion and temperature changes due to the superior superhydrophobicity, breathability, photo-thermal and electro-thermal effects, strain and temperature sensing performance. An example of using the Ti_3_C_2_T_x_ MXene/polyurethane composite fibers in the one-piece elbow sleeve, prepared by a commercial-scale flat-bed knitting machine, for tracking different movements through strain sensing and sending the signal by Bluetooth to a personal computer is presented in Figure 30d,e [198]. This single jersey knit of MXene/polyurethane four-ply yarn could recognize strains up to 200% and maintain stable work during 1000 stretching deformation cycles. 

3D printing can also be employed as a strategy to synthesize ideal flexible and multifunctional textiles. Cao et al. [197], for the first time, reported on the fabrication of a flexible smart textile with hybrid composite inks of TEMPO (2,2,6,6-tetramethylpiperidine-1-oxylradical)-mediated oxidized cellulose nanofibrils (TOCNFs) and Ti_3_C_2_ MXene. They prepared the Ti_3_C_2_ nanosheets from Ti_3_AlC_2_ powder by selective etching in LiF and HCl. The inks for 3D printing were obtained by mixing dispersions of TOCNFs and MXenes and magnetic stirring for 12 h at room temperature. These inks then could be transformed to continuous and stable gel TOCNFs/Ti_3_C_2_ fibers through extruding and injecting to an ethanol coagulation bath from a narrow nozzle of a 3D printer within seconds. Since these fibers possess good flexibility, they could be easily knotted after drying without causing any damage. The TOCNFs/Ti_3_C_2_ composite fibers also demonstrated enhanced mechanical properties. That was due to the numerous functional surface groups of MXenes providing many hydrogen bonds between the Ti_3_C_2_ and TOCNFs. Thus, breaking such fibers would need a large amount of energy. The TOCNFs/Ti_3_C_2_ composite fibers and textiles showed remarkable properties through photothermal and electrothermal functions in response to multiple external photon, electron, or strain stimuli.

MXene-based textiles, apart from the above-described functions, can also provide the EMI shielding effect [199]. The electronics industry is developing rapidly as electronic devices have become an integral part of medicine, communications, computations, space, and automation. The numerous systems located close to each other cause electromagnetic interference issues [200]. EMI occurs in the radio frequency range of the electromagnetic spectrum within 10^4^–10^12^ Hz [201]. Computers, fluorescent lamps, radio transmitters, and electric motors mainly radiate in the microwave range (1–40 GHz). EMI is suspected to be harmful to the human body [202,203]. Therefore, humans should be protected from the possible adverse effects of electromagnetic radiation. The way to control the influence of everyday exposure to EMI pollution is by using smart textiles capable of effective shielding [204].

EMI shielding materials first have been represented by metal-based foils, nanowires, and fibers from copper, nickel, stainless steel fiber, etc. However, the fact that such materials are heavy and susceptible to corrosion contributed to the emergence of a new class of EMI shielding materials: carbon-based composites and foams. Nevertheless, at low thickness, their protective abilities were found to be limited [205]. Therefore, the latest trend is introducing 2D MXenes into various types of fibers by different methods to achieve superior shielding characteristics [206]. The high electrical conductivity of MXenes enables the material with excellent shielding of electromagnetic waves [207].

For instance, high-elasticity and abrasion-resistant polyethylene terephthalate (PET) textiles that were dipped intoTi_3_C_2_T_x_ MXene solution demonstrated superb EMI shielding properties [187]. When MXene content was about 17.2 wt.%, the modified textile showed shielding efficiency of 42.1 dB in the X-band at a small thickness of 340 μm. The primary mechanism of EMI shielding in this textile was an absorption-dominant one. This is favorable since, in such a way, the secondary EMI pollution from reflection can be significantly reduced. Figure 31a shows a schematic representation of the MXene-decorated textile’s functioning. First, after the incident electromagnetic wave (EMW) strikes its surface, the high electrical conductivity of MXene nanosheets contributes to the reflection of a small part of the EMW. The rest EMW that entered the textile is trapped in the MXene nanosheets and is absorbed or dissipated in the form of heat within the material (textile itself has weaving and a porous structure providing an interface for effective reflection). If some part of the incident EMW is left, the multiple reflections are further absorbed and dissipated as MXene nanosheets have a large surface and interface area. The results of the electromagnetic radiation test, confirming the effectiveness of such a mechanism of EMI shielding in Ti_3_C_2_T_x_/PET textile, are presented in Figure 31b,c. The electromagnetic radiation tester was used to record the radiation from a far-infrared radiation (FIR) therapeutic lamp and cellphone before and after covering in Ti_3_C_2_T_x_/PET textile. As it can be seen, without the textile, the tester demonstrated 3703, 1378, and 433 µW cm^−2^ depending on the distance from the lamp. However, after using the MXene-decorated textile, the tester showed 0 µW cm^−2^ regardless of the distance. The same behavior was observed with a cellphone that generates severe electromagnetic radiation. The EM radiation was reduced from 4989 to 0 µW cm^−2^ by simply wrapping the cellphone.

The multifunctional MXene-decorated textiles are very helpful since they can be used for heating to keep warmth, protect from electromagnetic radiation generated from daily life, and sense human movements and temperature while being stretchable, breathable, and flexible. Nevertheless, assembling microscopic MXene nanosheets into macroscopic textiles can be challenging due to the irregular shape and size of the exfoliated MXenes. Many papers are devoted to Ti_3_C_2_T_x_ MXene, but other members of a big family are still not discovered for smart textiles for wearable multifunctional electronics applications. The obtained results are promising, and there are plenty of effects to be explained. MXenes have remarkable mechanical properties with an ultra-high elastic modulus and excellent electrical conductivity that makes them the leading materials for smart fibers and textiles so far [208].

## 5. Conclusions

The decade after their discovery, around 30 stoichiometric MXene family representatives have already been experimentally made and more than 100 are theoretically predicted. In addition, more than 20 solid solutions and high-entropy MXenes have been reported. Despite this fact, the majority of research focused on the first MXene family representative, Ti_3_C_2_, which has found application in many fields, including energy storage, photocatalysis, sensors, water purification, electrodes (including flexible electronics), as a SERS substrate, tribology, and biomedicine. Due to its outstanding electrical and optical properties, hydrophilic nature, and active surface area, several other MXenes representatives, Nb_2_C, Mo_2_C, Ta_4_C_3_, Mo_2_TiC_2_, etc., have been actively explored for various application during the last 5 years. 

Despite the great progress in MXene applications, some challenging issues need to be resolved before new MXene-based products head toward real markets. The first technological problem is in the use of primarily fluorine-containing acidic reagents, which are toxic for the biological environment, for MXene synthesis. Bottom-up synthesis may partially solve this issue but needs technological scale-up and improvement. Only first steps have been made in this direction. Molten salt etching is promising, if efficient delamination of salt-etched MXenes can be achieved. The price of MXene significantly decreased during the last years but it is still high, largely because the materials are still not available for mass production. MXenes may oxidize during the storage. This may significantly influence their properties with additional degradation over some application conditions. However, improved stoichiometry and flake perfection make MXenes stable for months in suspension and for many years in dry films. In recent years, new technologies and solutions applied have letdto significant improvement of MXene synthesis and post-production treatments that have greatly improved properties and opened wide perspectives for MXene applications.

## Figures and Tables

**Figure 1 nanomaterials-11-03412-f001:**
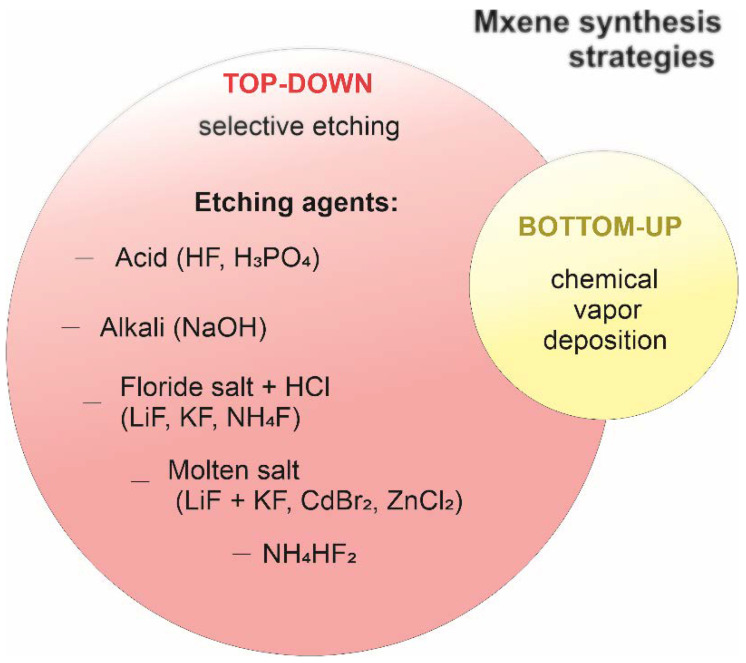
Methods for obtaining MXenes and examples of reagents used for selective etching of MAX phases or other precursor materials.

**Figure 2 nanomaterials-11-03412-f002:**
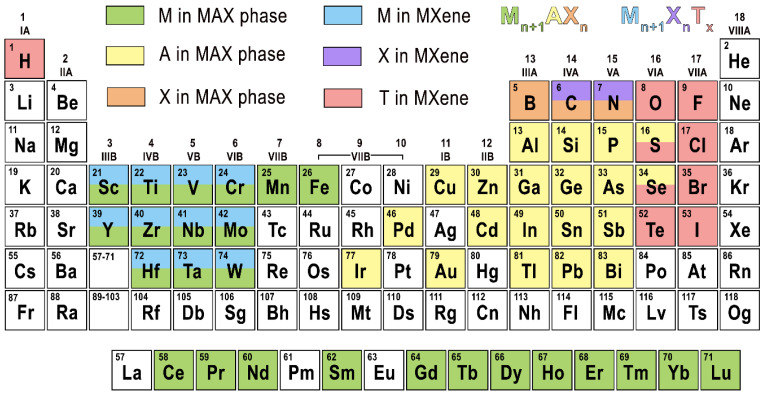
Elements of the known MAX phases and MXenes. A green background indicates the metals (M elements) that are used only to obtain MAX phases, but no MXene has been yet synthesized with them; the elements in M positions that have been successfully used for the synthesis of MXenes are highlighted in blue. The known elements A of the MAX phase layers are marked with a yellow background. X elements in MAX phases and MXenes are highlighted in orange and purple, respectively. Red background shows functional surface groups in MXenes.

**Figure 3 nanomaterials-11-03412-f003:**
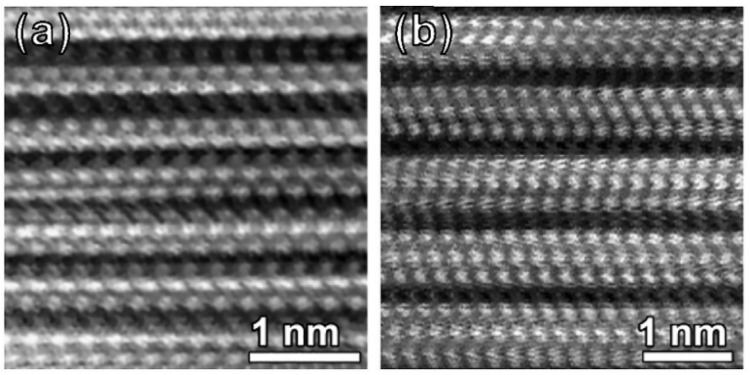
High-resolution Z contrast images of (V_0.5_Cr_0.5_)_2_AlC (**a**); (V_0.5_Cr_0.5_)_4_AlC_3_ (**b**) and (V_0.5_Cr_0.5_)_5_Al_2_C_3_ (**c**) in the direction [1$\bar{2}$10]. Reprinted with permission from [30] 2008 John Wiley and Sons.

**Figure 4 nanomaterials-11-03412-f004:**
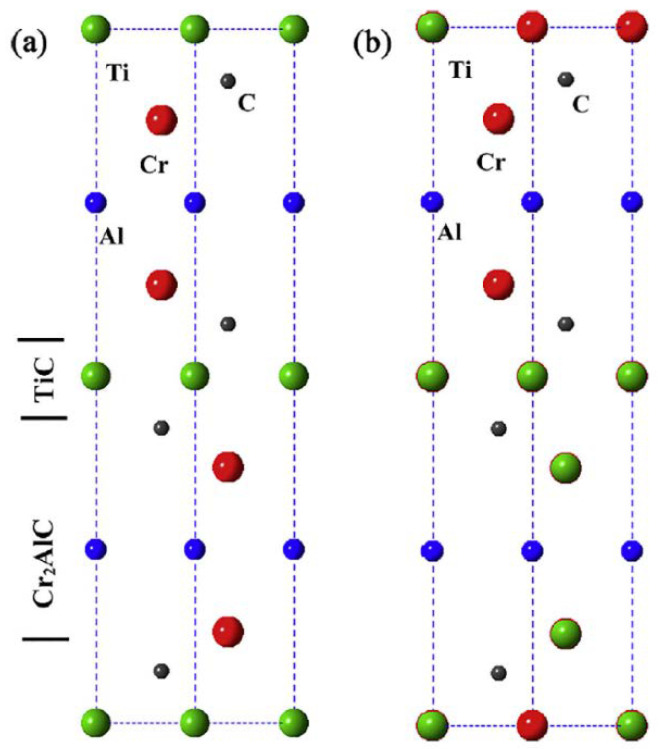
Crystal structure of (Cr_2/3_Ti_1/3_)_3_AlC_2_ with ordered M-positions (**a**) and randomly arranged atoms in M positions (**b**), projected onto the plane [1$\bar{2}$10]. Reprinted with permission from [32] 2014 Elsevier.

**Figure 5 nanomaterials-11-03412-f005:**
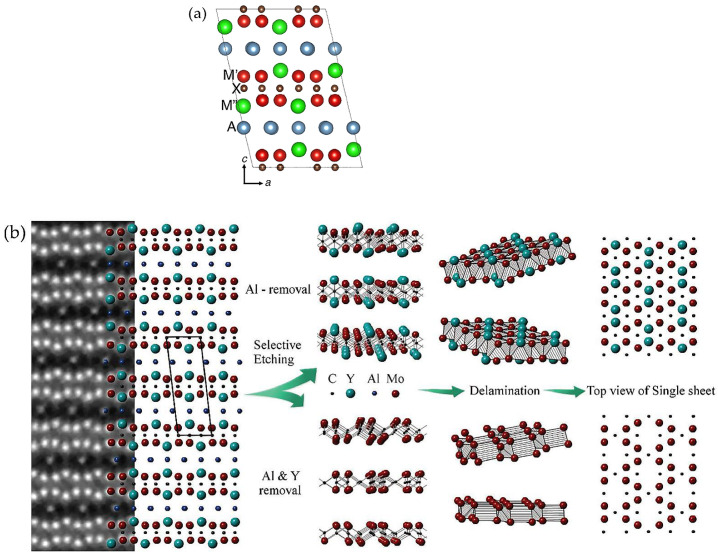
A side view of the crystal lattice cell M^1^_2/3_M^2^_1/3_)_2_AX *i*-MAX phase (**a**); the process of obtaining two types of Mxenes from the i-MAX phases (Mo_2/3_Y_1/3_)_2_AlC–(M^1^_2/3_M^2^_1/3_)_2_X and M_1,33_C (**b**). (Left-to-right) STEM image and schematic representation of the arrangement of phase 211 atoms along the [110] axis prior to etching. Two different structures were obtained by using two protocols for etching the MAX phase: removing only Al (top) or Al + Y (bottom). Top view of a single layer of synthesized Mxene (right). Obtained top view of the corresponding structures. Reprinted with permission from [29,42] 2018 American Physical Society and 2018 John Wiley and Sons.

**Figure 6 nanomaterials-11-03412-f006:**
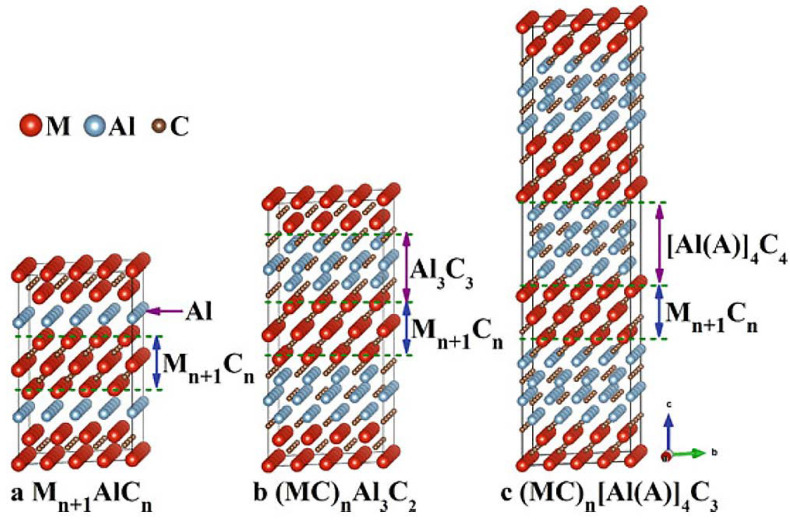
A comparison of the structure of the MAX-phase (M_n+1_AlC_n_) (**a**) and layered ternary (**b**) and quaternary (**c**) carbides of the type (MC)_n_[Al(A)]_m_C_m-1_ (n = 1, 2, 3, m = 3 and 4). Reprinted with permission from [49] 2019 Springer Nature.

**Figure 7 nanomaterials-11-03412-f007:**
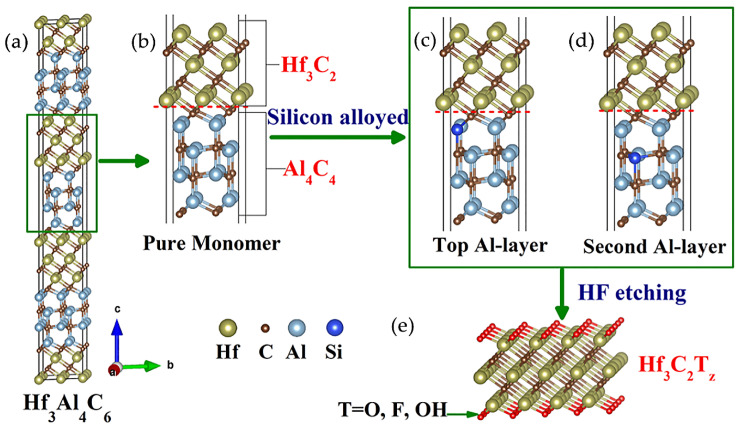
Schematic representation of the synthesis of the Hf_3_C_2_T_x_ MXene: (**a**) Hf_3_Al_4_C_6_ supercell without Si; (**b**) monomer Hf_3_Al_4_C_6_. The red dashed line shows the Al_4_C_4_ selective etch interface; (**c**,**d**) possible variants of the arrangement of Si atoms upon doping of the Al_4_C_4_ sublayer; (**e**) side view of synthesized MXene Hf_3_C_2_T_x_. Reprinted with permission from [51] 2017 American Chemical Society.

**Figure 8 nanomaterials-11-03412-f008:**
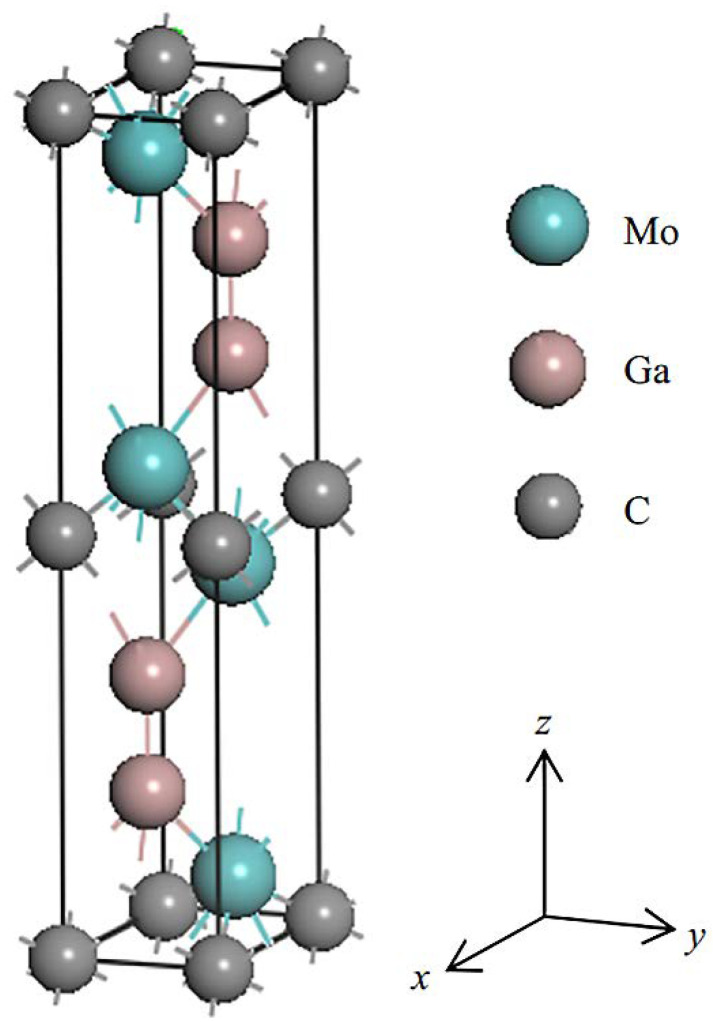
The unit cell of nanolaminated Mo_2_Ga_2_C carbide. Reprinted with permission from [52] 2016 Elsevier.

**Figure 9 nanomaterials-11-03412-f009:**
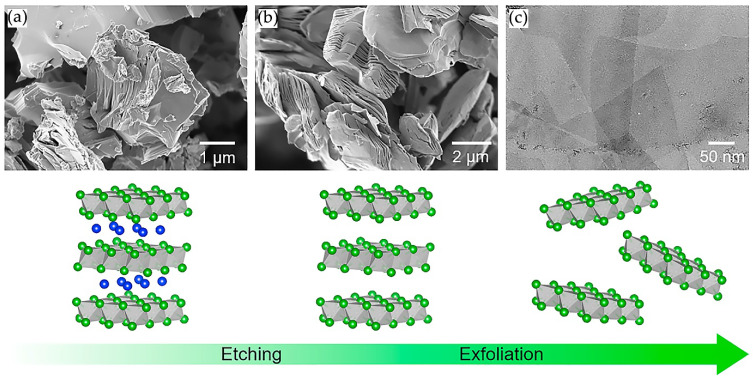
The process of synthesis of MXenes. (**a**), a three-dimensional precursor, which is used to etch certain elements; (**b**) a multilayer MXene obtained after etching; (**c**) stratification into single nanolayers. Reprinted with permission from [61] 2019 Elsevier.

**Figure 10 nanomaterials-11-03412-f010:**
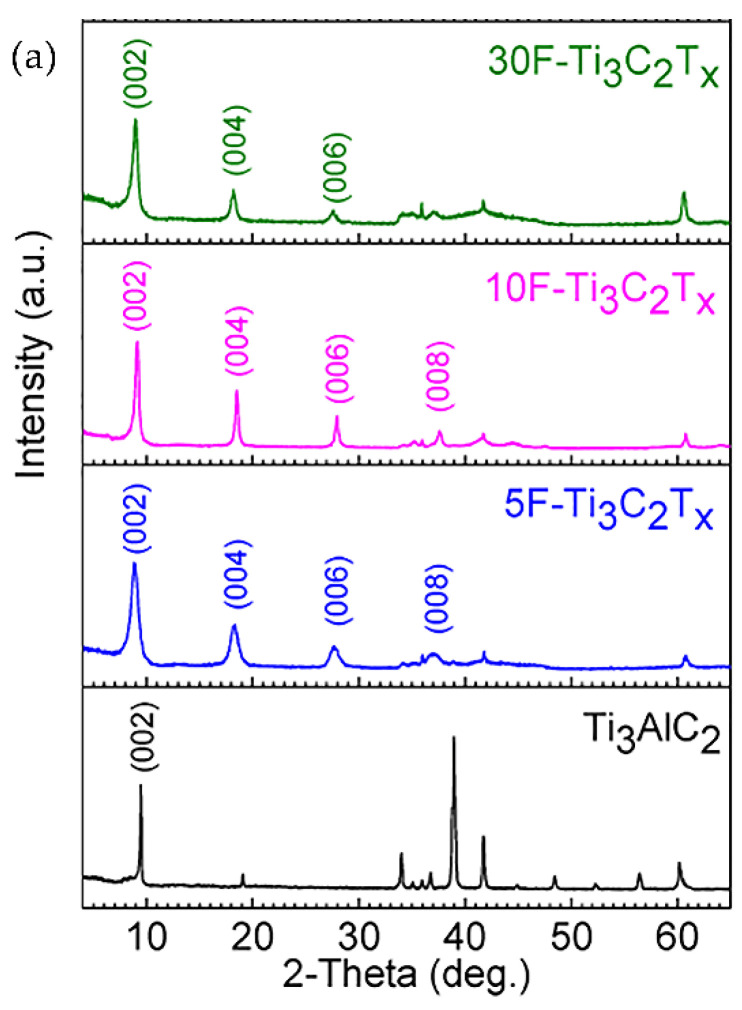
(**a**), X-ray spectra and SEM images of Ti_3_AlC_2_ MAX phase (**b**) and Ti_3_C_2_T_x_ MXene powders synthesized using (**c**) 5, (**d**) 10, and (**e**) 30 wt.% HF. Reprinted with permission from [63] 2017 American Chemical Society.

**Figure 11 nanomaterials-11-03412-f011:**
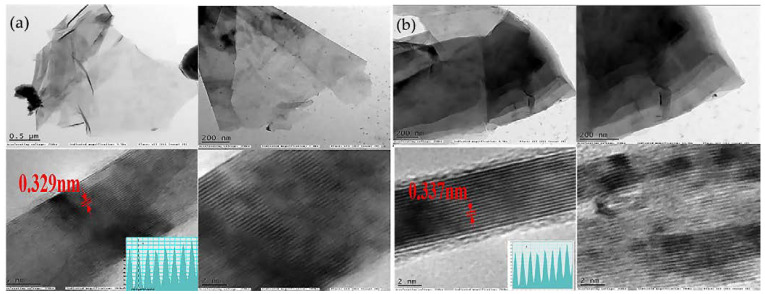
TEM micrographs of (**a**) Ti_x_Ta_(4−x)_AlC_3_ MAX phases and (**b**) Ti_x_Ta_(4−x)_C_3_ MXene. Reprinted with permission from [64].

**Figure 12 nanomaterials-11-03412-f012:**
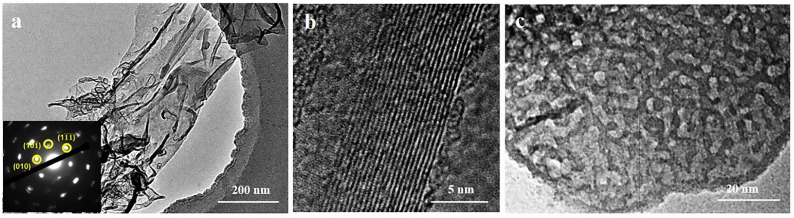
High-resolution transmission electron microscope (HRTEM) images of Mo_2_C MXene, showing (**a**,**b**) a layered structure and (**c**) a mesoporous surface. Reprinted with permission from [37] 2020 Elsevier.

**Figure 13 nanomaterials-11-03412-f013:**
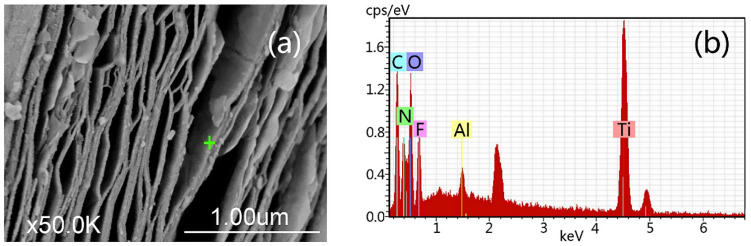
(**a**), A high-resolution SEM image of morphology of the Ti_3_C_2_T_x_ MXene synthesized using NH_4_F and (**b**) chemical composition studied by energy dispersive spectroscopy (EDS). Reprinted with permission from [66] 2016 Springer Nature.

**Figure 14 nanomaterials-11-03412-f014:**
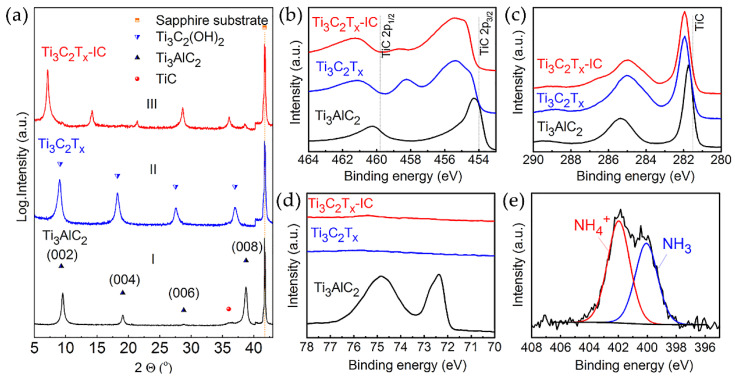
(**a**) X-ray spectra of (I) Ti_3_AlC_2_ 60 nm thick precursor film, Ti_3_C_2_T_x_ MXene after etching in (II) 50% HF for 160 min and (III) 1 M NH_4_HF_2_ for 660 min. The XPS spectra are shown in the right panels, where (**b**) Ti 2p, (**c**) C 1s, (**d**) Al 2p, and (**e**) N 1s (for MXene synthesized with NH_4_HF_2_), where two components correspond to NH_4_^+1^ and NH_3_ [62].

**Figure 15 nanomaterials-11-03412-f015:**
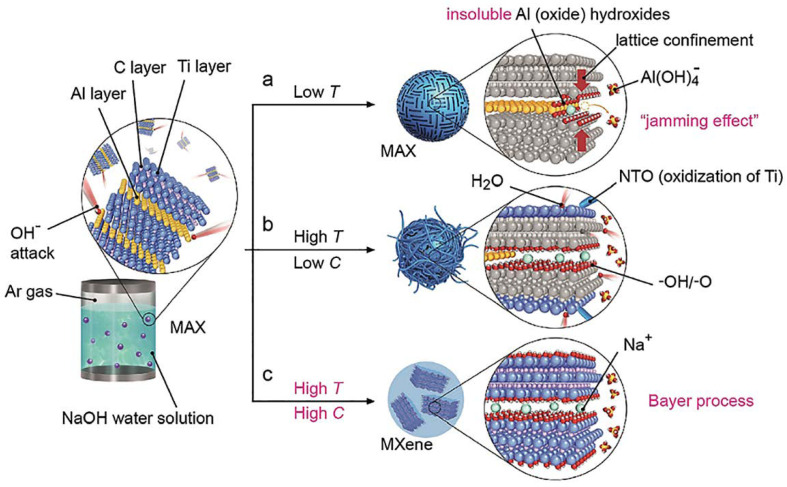
Schematic representation of the reaction between Ti_3_AlC_2_ MAX-phase and NaOH under different conditions: (**a**) aluminum hydroxides (oxides) block the Al etching process at low temperatures; (**b**) some aluminum hydroxides (oxides) dissolve in NaOH at high temperatures and low NaOH concentrations, but high water content leads to oxidation of MXene and the formation of Na/K-Ti–O compounds; (**c**) high temperatures and high NaOH concentrations lead to the dissolution of aluminum hydroxides (oxides) in an alkali solution. Reprinted with permission from [73] 2018 John Wiley and Sons.

**Figure 16 nanomaterials-11-03412-f016:**
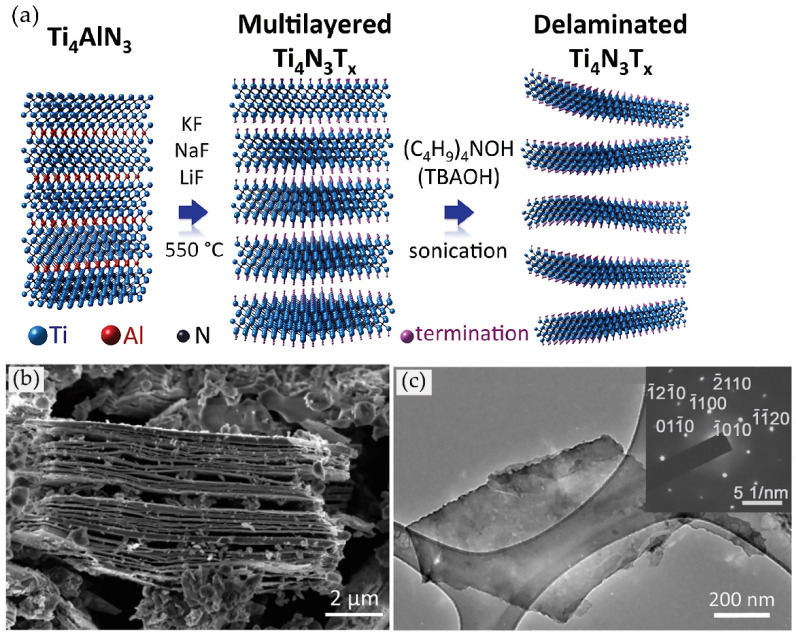
(**a**) Schematic representation of the process of obtaining the first nitride MXene by treatment of Ti_4_AlN_3_ with a mixture of molten fluoride salts at 550 °C in Ar atmosphere, followed by delamination of multilayer Ti_4_N_3_T_x_ using TBAOH. (**b**) SEM image of multilayer MXene after etching; and (**c**) TEM micrograph of a separate Ti_4_N_3_T_x_ layer with electron diffraction from a selected region, which demonstrates the hexagonal symmetry of the basal plane. Reprinted with permission from [75] 2016 RSC Pub.

**Figure 17 nanomaterials-11-03412-f017:**
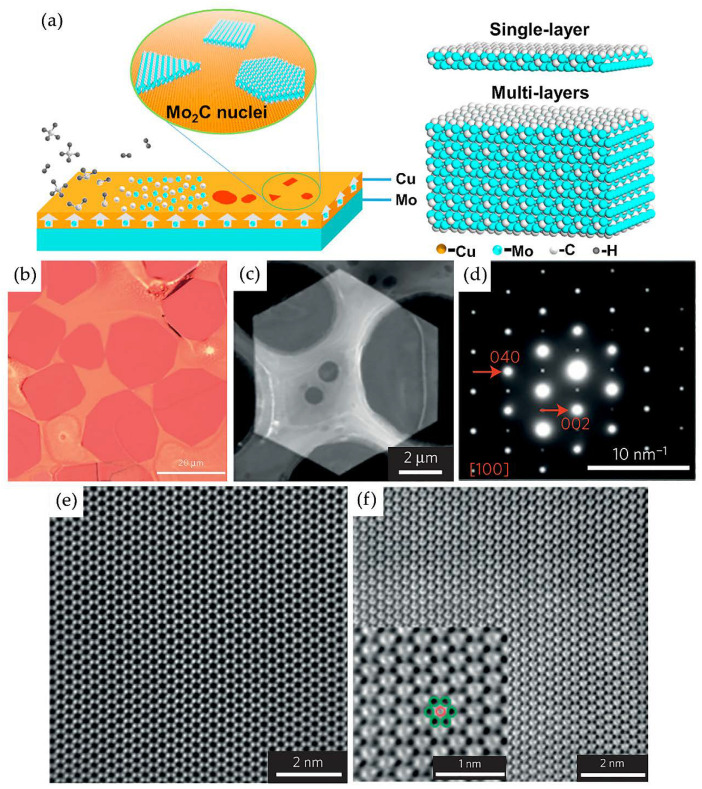
(**a**) Schematic representation of the synthesis of two-dimensional Mo_2_C crystals on a Cu/Mo substrate and a model of the atomic structure depending on their thickness. Images of ultrathin α-Mo_2_C films with different shapes (optical microscope). STEM images of hexagonal α-Mo2C obtained with a high-angle annular darkfield detector with low (**b**) and high (**c**) resolution. (**d**) Electron diffraction pattern of the selected area along the [100] axis. Atomic resolution high-angle annular dark-field (HAADF)-STEM and (**f**) bright-field (BF)-STEM micrographs. Mo atoms, which are seen as white dots € (**e**), are located in a hexagonal close-packed structure, while C atoms are centered between six Mo atoms. Green circles in (**f**) indicate Mo atoms, and the red circle is assigned to C. Reprinted with permission from [77,78] 2019 Springer Nature and 2015 Springer Nature.

**Figure 18 nanomaterials-11-03412-f018:**
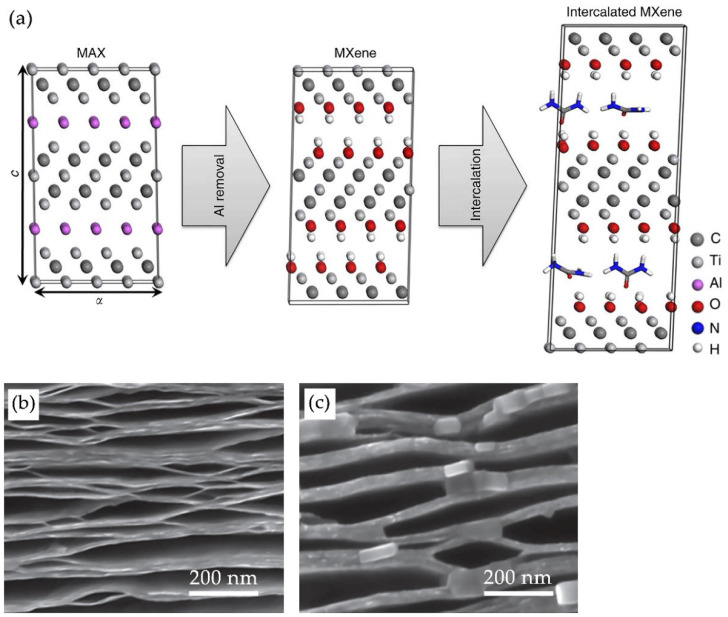
(**a**) Scheme of the synthesis of Ti_3_C_2_T_x_ MXene with selective etching of the MAX phase and subsequent intercalation (for example, urea is shown as an intercalant). SEM micrographs of Ti_3_C_2_T_x_ (**b**) before and (**c**) after intercalation with hydrazine monohydrate dissolved in N,N-dimethylformamide for 24 h at 80 °C. Reprinted with permission from [9] 2013 Springer Nature.

**Figure 19 nanomaterials-11-03412-f019:**
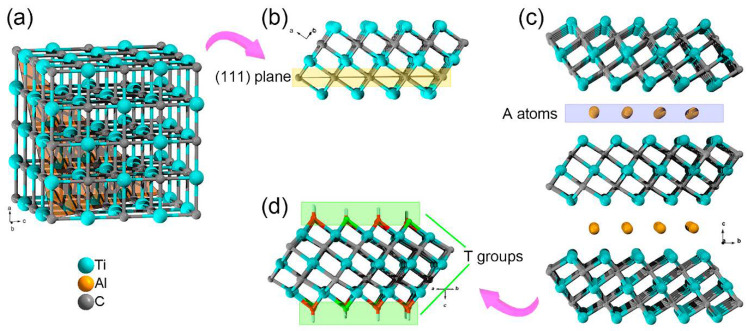
(**a**) Typical fcc lattice of transition metal carbide M; (**b**) a layer separated from the fcc lattice along the (111) plane; (**c**) side view of M_3_AX_2_, and (**d**) MXene monolayer M_3_X_2_T_x_ with a terminated surface [100].

**Figure 20 nanomaterials-11-03412-f020:**
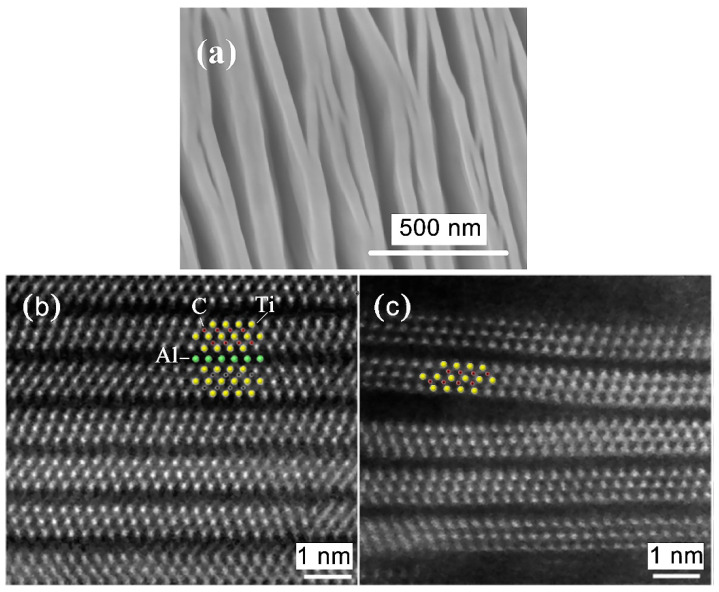
(**a**) SEM image of Ti_3_C_2_ after removing the A layer; high-resolution STEM images of the MAX phase Ti_3_AlC_2_ (**b**) and MXene Ti_3_C_2_T_x_ (**c**) along the [$11\bar{2}0$] axis. The Ti, C, and Al atoms are highlighted in yellow, red, and green circles. Adapted from [62]. Reprinted with permission from [103] 2021 Elsevier.

**Figure 21 nanomaterials-11-03412-f021:**
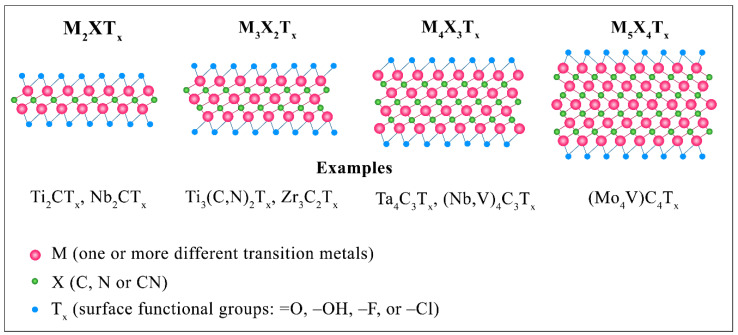
Schematic representation of the structure of MXenes with n = 1–4 and some examples.

**Figure 22 nanomaterials-11-03412-f022:**
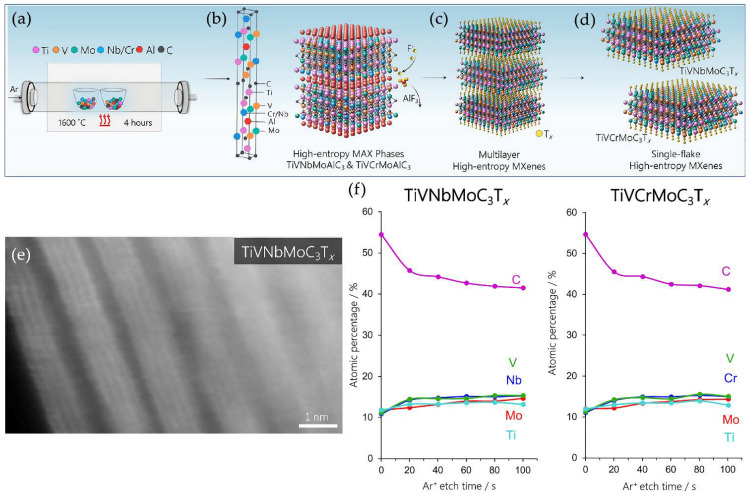
Schematic illustrations of the (**a**) synthesis of high-entropy alloy MAX phases through sintering of powers with stoichiometric molar ratios, (**b**) unit cell of the TiVNbMoAlC_3_ and TiVCrMoAlC_3_ (M_1_M_2_M_3_M_4_AlC_3_) MAX phases, where elements are designated as follows: Ti—pink, Nb/Cr—blue, Al—red, V—orange, Mo—green, and C—black. Selective etching of the Al layers by hydrofluoric acid to synthesize multilayer high-entropy MXenes. The process of the selective etching of the MAX phase for MXene synthesis. (**c**) Multilayered and (**d**) single flakes of high-entropy MXenes TiVNbMoC_3_T_x_ and TiVCrMoC_3_T_x_ after delamination, where functional surface groups (-O/-F) are colored in yellow. (**e**) SEM image of the high-entropy TiVNbMoC_3_T_x_ MXene, showing four layers of metals, which is characteristic of the M_4_C_3_T_x_ structure. (**f**) XPS in-depth distribution of metal elements and carbon in synthesized MXenes. Reprinted with permission from [104] 2021 American Chemical Society.

**Figure 23 nanomaterials-11-03412-f023:**
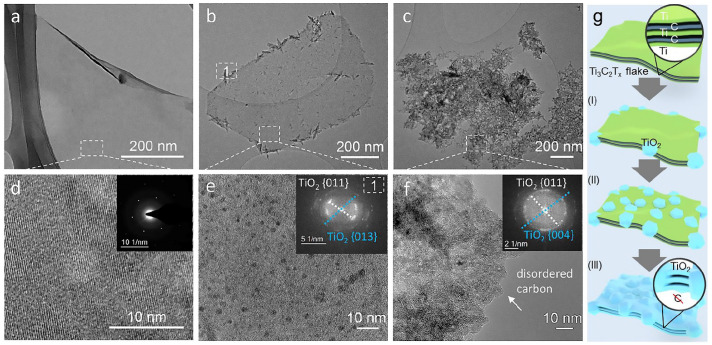
TEM images of Ti_3_C_2_T_x_ MXene nanolayers (**a**) immediately after synthesis and kept in the air at room temperature for (**b**) 7 days and (**c**) 30 days. (**d**–**f**) High-resolution TEM images corresponding to (**a**–**c**). Inserts in (**d**) are the electron diffraction pattern of the selected areas, and (**e**,**f**) is the spatial fast Fourier transform of the images. (**g**) Diagram of the degradation process of a Ti_3_C_2_T_x_ nanolayer: Stage I—formation of TiO_2_ at the MXene edge, Stage II—growth of TiO_2_ nanoparticles from the edges towards the surface while maintaining the carbon layer, Stage III—etching of carbon by hydroxyl radicals generated by the surrounding TiO_2_, followed by complete transformation MXene to amorphous titanium oxide. Reprinted with permission from [112,113] 2017 American Chemical Society and 2020 American Chemical Society.

**Figure 24 nanomaterials-11-03412-f024:**
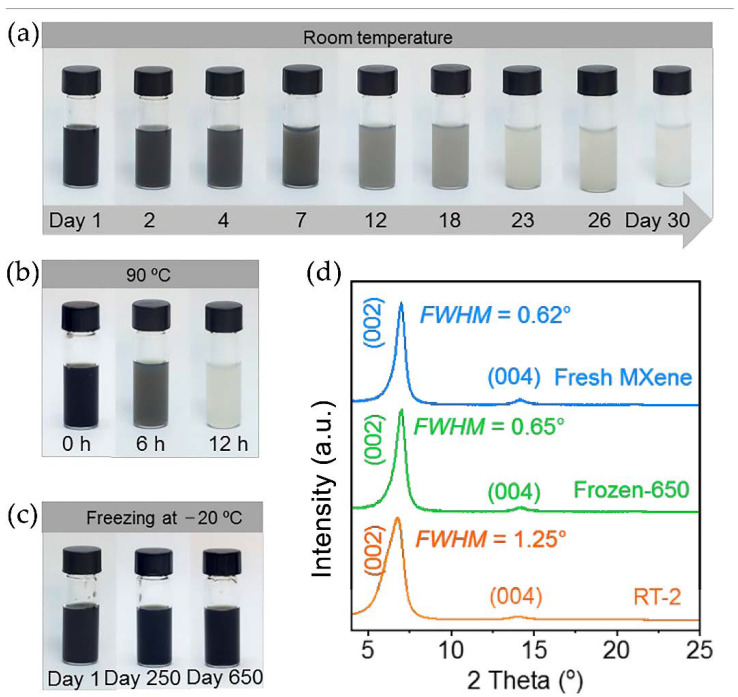
Photographs of an aqueous dispersion of Ti_3_C_2_T_x_ (0.5 mg/ml) showing the effect of temperature on the oxidation rate of MXene: (**a**) color change when the aqueous dispersion is stored at room temperature for 30 days, (**b**) at 90 °C for 0, 6, and 12 h, and (**c**) at −20 °C for 250 and 650 days. (**d**) X-ray diffraction patterns of MXene films obtained from Mxene dispersions immediately after synthesis, frozen at −20 °C after 650 days of storage and Mxene held for 2 days at room temperature. Reprinted with permission from [113] 2020 American Chemical Society.

**Figure 25 nanomaterials-11-03412-f025:**
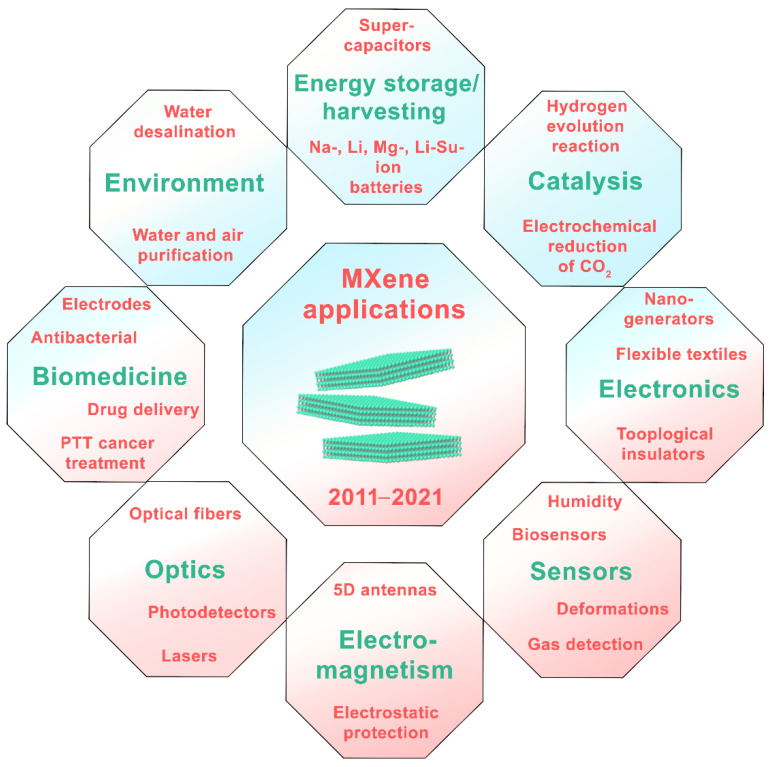
Possible applications of two-dimensional MXene in various industries.

**Figure 26 nanomaterials-11-03412-f026:**
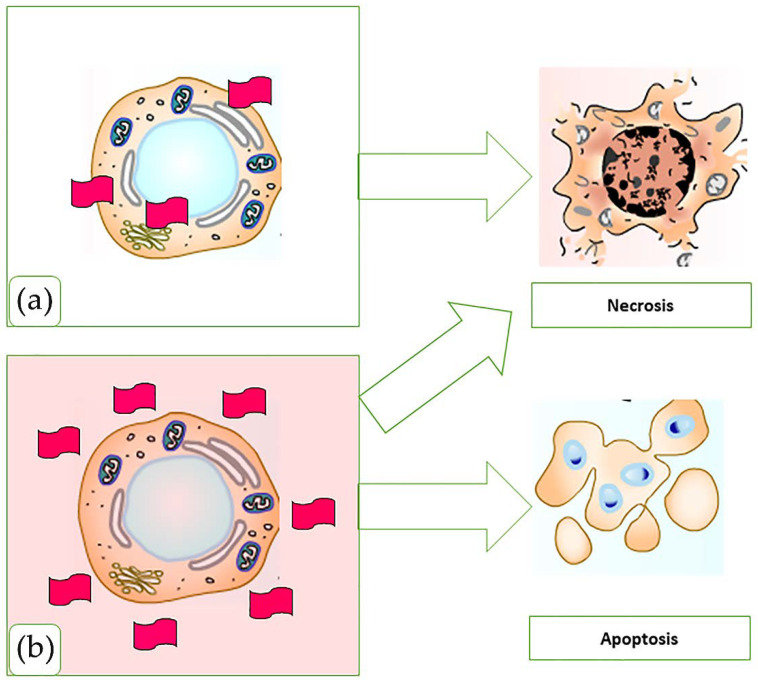
Proposed mechanisms of anticancer effects of MXenes (figure description provided in the main text).

**Figure 27 nanomaterials-11-03412-f027:**
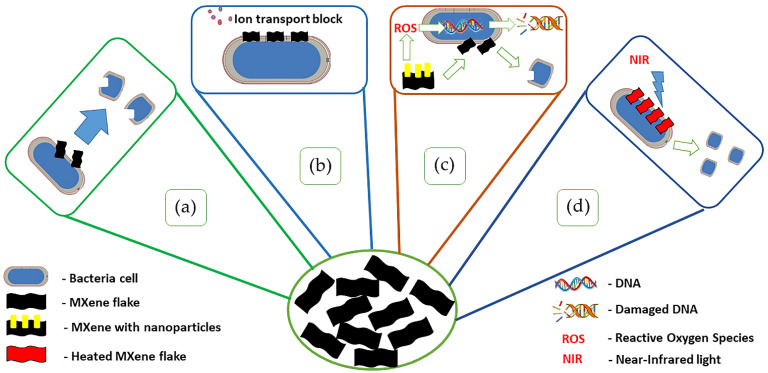
The scheme of proposed mechanisms of MXene antibacterial activity. (**a**)—a direct invasion of sharp MXene to bacteria wall with cell membrane disintegration; (**b**)—formation of the conductive bridge over the insulating lipid bilayer and ion transport block; (**c**)—MXene-AuNCs provide direct MXene invasion to bacteria cells with formation of ROS by AuNCs and DNA damage; (**d**)—photo-thermal ablation of bacteria after NIR light irradiation of MXenes.

**Figure 28 nanomaterials-11-03412-f028:**
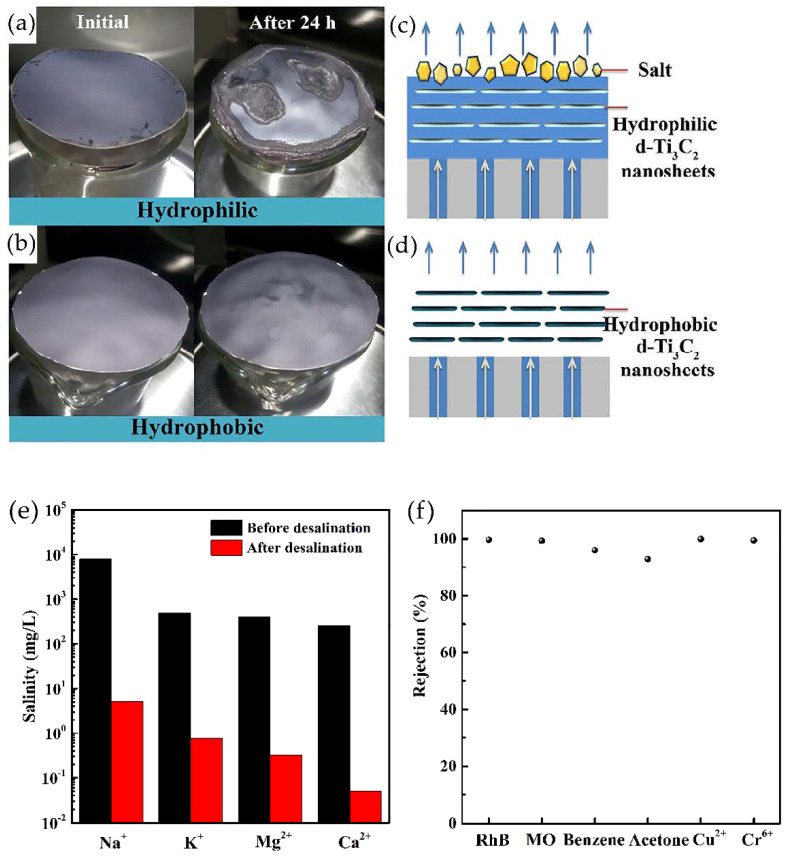
Optical photographs of (**a**) hydrophilic and (**b**) hydrophobic membranes with Ti_3_C_2_ before and after 24 h of desalination. Schematic representation of the desalination process by solar steam generation using (**c**) hydrophilic and (**d**) hydrophobic membranes. (**e**) Results of measuring the salinity of Na^+^, K^+^, Mg^+^, and Ca^+^ primary ions before and after desalination. (**f**) Filtration efficiency of organic substances and heavy metal ions. Reprinted with permission from [167] 2018 Royal Society of Chemistry.

**Figure 29 nanomaterials-11-03412-f029:**
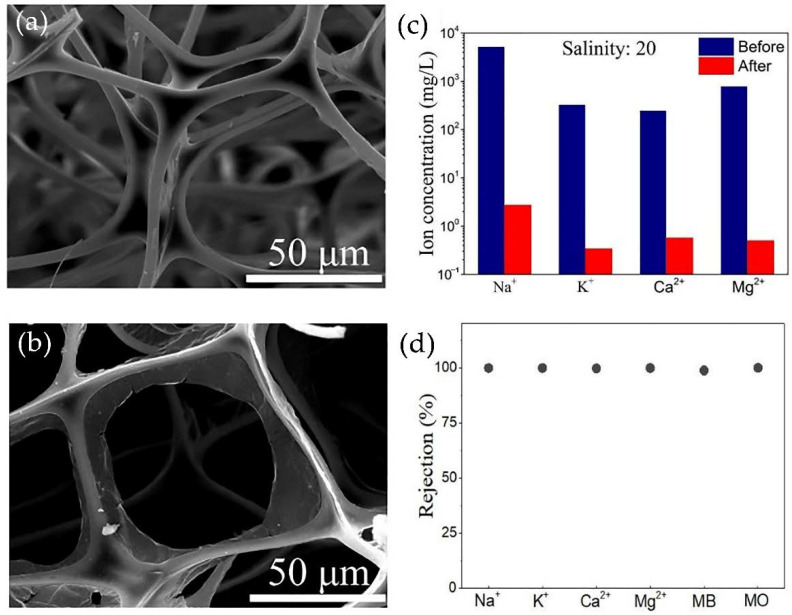
SEM images of melamine foam (**a**) before and (**b**) after immersion in a solution of polyvinyl alcohol and MXene suspension, followed by drying. (**c**) Concentrations of Na^+^, K^+^, Mg^+^, and Ca^+^ cations in a standard seawater sample with a salinity of 20 before and after evaporation. (**d**) Degree of trapping of metal ions and dyes (methylene blue, MS, and methyl orange, MO). Reprinted with permission from [165] 2019 Royal Society of Chemistry.

**Figure 30 nanomaterials-11-03412-f030:**
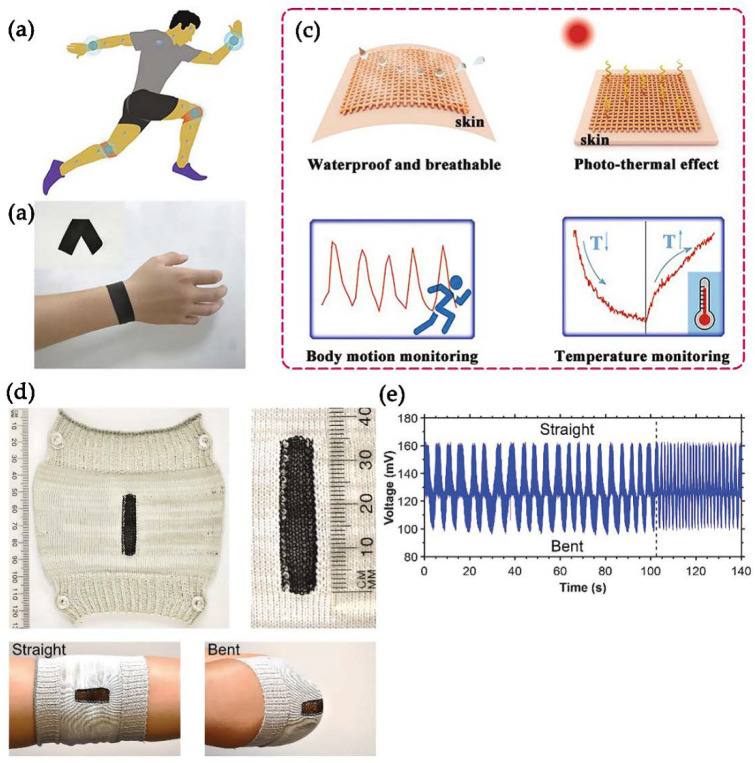
(**a**) Schematic illustration of the PM/PDMS textile attached to the joints of the human body to monitor motion actions. (**b**) A photograph of the volunteer wrist with MXene-based textile serving as a smart strain sensing device; the inset shows the photograph of one black ring textile. (**c**) Promising applications and superior properties. Reprinted with permission from [196] 2021 Elsevier. (**d**) One-piece sleeve for elbow knitted using four-ply yarn of Ti_3_C_2_T_x_ MXene/polyurethane fiber in as-prepared and functioning states. (**e**) Strain sensing response of the elbow sleeve collected during continuous bending and straightening the elbow at two different frequencies (0.2 and 0.7 Hz) through a wireless Bluetooth connection. Reprinted with permission from [198] 2020 John Wiley and Sons.

**Figure 31 nanomaterials-11-03412-f031:**
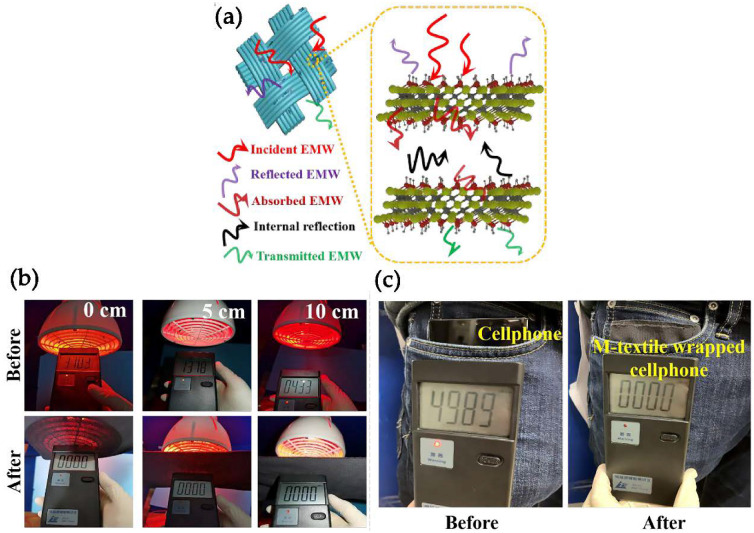
(**a**) Schematic of electromagnetic microwave dissipation in the MXene-decorated textiles. A electromagnetic radiation tester used to check the radiation before and after applying Ti_3_C_2_T_x_/PET textile on a FIR lamp (**b**) and a cellphone (**c**). Reprinted with permission from [187] 2020 Royal Society of Chemistry.

**Table 1 nanomaterials-11-03412-t001:** A list of various MXenes known to date with their features and possible applications.

№	MXene	Synthesis	Specific Features	Applications	Refs
1	Mo_2_C	Selective etching of Ga atoms from Mo_2_Ga_2_C using: (1) hydrofluoric acid (HF); (2) a solution of LiF and HCl; (3) ultraviolet radiation and phosphate acid; (4) a solution of NH_4_F with hydrochloric acid (HCl) under hydrothermal conditions for 24 h at 140–180 °C; magnetron sputtering	High activity of the electrochemical reaction of hydrogen evolution, since the basal planes of Mo_2_CT_x_ are catalytically active with respect to the reaction of hydrogen evolution; In the air, Mo_2_C is stable at 200 °C, and at 590 °C, it is completely oxidized to MoO_3_; Adsorbed COOH groups can spontaneously dissociate into a water molecule and CO, adsorbed on Mo_2_C; Flexible batteries (bending angle of about 110 °) based on Mo_2_C have excellent capacity retention of ~89% and ~74% for lithium-ion and sodium-ion batteries, respectively	Electrocatalysts for the evolution of hydrogen; high-performance flexible energy storage devices; electrocatalyst for the reduction of CO_2_; saturable absorber for passively Q-switched lasers	[37,80,89,90,91]
2	Nb_2_C	Selective etching of Al atoms from Nb_2_AlC powder using 50% aqueous solution of hydrofluoric acid	High biocompatibility; High efficiency of photothermal conversion while maintaining the required photothermal stability; Capable of penetrating organelles through Matrigel; Demonstrates ultra-stable pulses in the telecommunications and mid-infrared regions; At a center wavelength of 1882 nm, the 69th order harmonic can be achieved from 411 MHz; Excellent electrocatalytic characteristics when terminating the surface with -O functional groups. Nb_2_C cathodes are stable for 130 cycles at an ultra-high current density of 3 A/g	Photothermal therapy of cancer; Protecting and stimulating the survival of intestinal cells during various procedures; Building block for narrow-band photoelectrochemical photodetectors and mode synchronizers; cathode material for lithium–oxygen batteries	[92,93,94,95]
3	Ti_3_C_2_	Selective etching of Al atoms from Ti_3_AlC_2_ using: (1) hydrofluoric acid; (2) sodium hydroxide solution (NaOH); (3) ammonium chloride; (4) a molten salt of ZnCl_2_ (the ratio of Ti_3_AlC_2_ and ZnCl_2_ was 1:6); NH_4_HF_2_- containing polar organic solvents (without water)	High biocompatibility; High electron affinity between thrombin and Ti_3_C_2_; High efficiency of fluorescence quenching; Ultra-high ability to remove the typical cytokine IL-6 through a mechanism of chemisorption	Photothermal therapy of cancer; Aptasensor based on Förster resonance energy transfer for the quantitative determination of thrombin; Hemoperfusion absorbent for blocking cytokine storm for treatment of severe COVID-19 infection; Anode material of Na-ion batteries	[72,76,92,96]
4	Ta_4_C_3_	Etching Al atoms from Ta_4_AlC_3_ with 40% aqueous solution of HF + solution of N-methylpyrrolidone	High biocompatibility; The efficiency of photothermal conversion is about 44.7%; Excellent photothermal stability; High photothermal performance in the near-infrared range	Photoacoustic computed tomography of tumors with contrast enhancement; In vivo photothermal ablation of tumor xenografts; Theranostics	[82]
5	Mo_2_TiC_2_	Etching of Al atoms from Mo_2_TiAlC_2_ with a 48–51% aqueous solution of HF	Shows properties of a semiconductor. Resistivity increases with decreasing temperature (*dρ/dT* < 0) in the measuring range 10–250 K	Applications in electronics and optics	[39]
6	Ti_x_Ta_4−x_C_3_	Hydrofluoric acid etching of Al atoms from Ti_x_Ta_4−x_AlC_3_	Stable electrochemical characteristics, high capacity, and good performance: reversible specific capacity of 459 mAh/g at 0.5 °C for 200 cycles with a capacity retention of about 97%; Bimetallic MXene accumulates ions on the surface of its layers	Anode material for lithium-ion batteries	[64]
7	(Mo_4_V)C_4_	Hydrofluoric acid etching of Al atoms from (MoV)_5_AlC_4_ (the ratio of Mo:V precursor powders was 4:1)	The presence of disordered M-positions; The structure is stable up to 900 °C with subsequent transformation into the orthorhombic (Mo, V) 2C phase, and at 1500 °C into cubic c (Mo, V) C; The specific electrical resistance of MXene is about 1.20 mΩ cm, and the conductivity was 833 S/cm). However, the resistance is worse compared to Mo2C and Mo2TiC2 (0.80 and 0.67 mΩ cm)	-	[97]
8	Ti_4_N_3_	Etching of Ti_4_AlN_3_ in a mixture of 59 wt.% KF + 29 wt.% LiF + 12 wt.% NaF at 550 °C with TBAOH	Ti_4_N_3_ with functional surface groups O, F, or OH has higher states at the Fermi level compared to similar two-dimensional carbides; Magnetic moment about 7.0 μB per unit cell	Electrodes in electrochemical capacitors; Plasmonic material for conversion optics	[75]
9	V_2_N	Etching of V_2_AlN with a LiF-HCl mixture, followed by treatment with TMAOH or DMSO separating agents	Outstanding electrochemical stability; Higher electronic conductivity than carbide MXene	Electrodes for supercapacitor	[70]
10	Ti_3_CN	Etching of Al atoms from Ti_3_AlCN powder in 30% HF solution	Abnormally high absorption of electromagnetic waves in the layered structure of carbonitride MXene after thermal annealing at 350 °C; Electrical conductivity at 2475 S/cm after annealing at 250 °C	Anode material for Na- and Li-ion batteries;Spectral filter that induces mode-locked laser pulses for photonic applications; Lightweight, ultra-thin, and flexible EMI shielding materials	[98,99]

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
