# Peer review of "MXenes—A New Class of Two-Dimensional Materials: Structure, Properties and Potential Applications"

_nanomaterials, 2021, doi:10.3390/nano11123412_

Round 1

Reviewer 1 Report

Report on the paper by Pogorielov et al.

It is an excellent review article on MXene – a new class of two-dimensional materials. Thus I would like to recommend the publication of the paper after minor changes.

  1. In line 292, it is required to describe the ultrasonic frequency and intensity (W/cm2) as well as type of sonication; horn or bath type.

  1. In line 378, it is required to describe the ultrasonic frequency and intensity (W/cm2) as well as type of sonication in the ultrasonic treatment; horn or bath type. It is also required to describe whether any other gases were dissolved in the oxygen-free deionized water. If no gas was dissolved in the water, it should be described as degassed water.

  1. With regard to the application of MXene to supercapacitor in Fig. 25, dielectric properties of MXene should be described in the manuscript such as the values of dielectric constant, its temperature dependence, and its frequency (of applied AC electric field) dependence.

Author Response

We thank the Reviewer for the careful revision of our manuscript. Your comments are very important and provide significant impact to manuscript quality. Below, we provide line-by-line answer to your comments

  1. In line 292, it is required to describe the ultrasonic frequency and intensity (W/cm2) as well as type of sonication; horn or bath type.

 In line 292 we describe the typical procedure of synthesizing MXenes, and that is why we do not give the certain sonication parameters. However, we consider that the comment is very valuable because it is a great idea to add some details about the sonication process. To describe it, we have added the following part to the text of the manuscript:  “Two types of ultrasonic devices can be employed for this purpose: bath and probe/tip sonication systems. The probe sonicators produce smaller flake sizes of delaminated MXenes. At the same time, bath sonication has been found to be more suitable for fabricating MXenes with larger flakes. The most critical parameters of the sonication process are amplitude, power, and frequency [54]. The irradiation amplitude significantly influences the intensity of ultrasonication. The high vibration amplitudes during the treatment provide higher energy to the solution, which negatively affects the stability of MXene due to the severe collapse of cavitation bubbles, collisions between delaminated MXenes and particles of the MAX phase. Thus, it is necessary to choose an optimal amplitude value and lower the concentration of particles in solution through dilution. The amount of power should also be tailored: it must be sufficient to generate cavitation. In contrast to the amplitude, higher frequencies positively impact the MXene stability. They cause a high number of small bubbles with uniform sizes, preventing violent interparticle collapses and promoting the weakening of the interlayer Van der Waals forces [55]. Papers report different frequencies: from 6 kHz [56] to 40 kHz [57–59]. However, the highest quality of MXene flakes was observed at sonication frequencies above 20 kHz, and the most commonly used one is 40 kHz.”

  1. In line 378, it is required to describe the ultrasonic frequency and intensity (W/cm2) as well as type of sonication in the ultrasonic treatment; horn or bath type. It is also required to describe whether any other gases were dissolved in the oxygen-free deionized water. If no gas was dissolved in the water, it should be described as degassed water.

Unfortunately, authors of the paper did not provide detailed parameters of the sonication process, only the information that the bath type system was used. The same is applied to the deionized water, the only description is that it was oxygen-free, no additional information about presence of other gases. Therefore, we could only add the phrase “using ultrasonic treatment in bath type sonication system…”

  1. With regard to the application of MXene to supercapacitor in Fig. 25, dielectric properties of MXene should be described in the manuscript such as the values of dielectric constant, its temperature dependence, and its frequency (of applied AC electric field) dependence.

Dear Reviewer! Thank you very much for this comment! Supercapacitors is one of the most important application of the MXenes and there are some review regarding this topic. The aim of current review was not cover all possible application but we add some new information in page 34: “The development of electrical power systems causes a need for new dielectrics, which must exhibit low dielectric loss and high dielectric constant to be suitable for electro-static film capacitors with high energy density, flexibility, and high breakdown strength. It was found that incorporation of MXene nanosheets to polymer allows achieving the optimal combination of high dielectric constant and low dielectric loss [119]. For instance, poly(vinylidene fluoride) (PVDF)-based percolative composites with 2D Ti3C2Tx nanosheets as fillers reached a dielectric constant as high as 105 near the percolation limit (15.3 wt.% MXene) [120]. Moreover, the dielectric loss of the MXene/P(VDF-TrFE-CFE) composite increased from 0.06 to 0.35 (5-fold), but the die-lectric constant increased by 25 times within 0 - 10 wt.% MXene composition range. The origin of the substantial permittivity enhancement is primarily due to the micro-scopic dipoles formed by the accumulation of charges at the interfaces between the MXene fillers and the polymer matrix. Compared with other fillers (hydrothermally reduced graphene oxide, copper phthalocyanine, functionalized graphene nanosheets), MXenes provided the best dielectric constant/loss factor trade-off. The other example of such composite is multilayered Ti3C2Tx/PVDF films fabricated by three steps: spin coating, spray coating, and hot-press methods [119]. This structure consisted of over-lapped layers of MXene and PVDF placed on each other to provide enhanced interfa-cial interactions due to Ti3C2Tx and good charge carrier insulation through the ferroe-lectric PVDF layer. The dielectric constant of the multilayer 4MXene/5PVDF (4 layers of Ti3C2Tx and 5 layers of PVDF) measured at 1 kHz was about 41, higher than that of pure PVDF (10.5). An increase was caused by enhanced Maxwell– Wagner–Sillars (MWS) interfacial polarization due to the difference in dielectric performances of Ti3C2Tx and PVDF. The dielectric loss at 1 MHz was suppressed below 0.2. Additionally, MXene/PVDF films had low conductivity of <109 S m-1 at 1 kHz. The frequency-dependent alternating current (AC) conductivity suggested an insulating behavior of the composite films. It also demonstrated a superior dielectric constant to loss factor ratio of about 1464.3. Therefore, composite MXene-based films are promising broad-band dielectric material for high-frequency capacitors. Other examples of the use of MXene phases as reinforcement in ceramic composites is Ti3C2Tx/Al2O3, which at 2 wt.% of Ti3C2Tx showed 300%, ~150%, ~300% improvement of the fracture toughness, bending strength, and hardness, respectively [121]. The ZnO-Ti3C2 composite produced by the cold sintering process improved the electrical conductivity of the oxide matrix by 1–2 orders of magnitude and showed a 150 % increase in hardness and elastic mod-ulus [122]. The enhancement of mechanical properties of silicon carbide modified with Ti3C2Tx was also reported by Petrus et al. [123]. The Ti3C2Tx MXene composite films with segregated polystyrene inclusions studied by Iqbal et al. [124] showed superior electromagnetic interference efficiency, making them a promising shielding material with tunable electromagnetic wave absorption properties. Thus, MXenes are popular materials with many possible applications in various industries. The Some of them main applications of MXene in various industries are schematically shown schemati-cally in Figure 25.”

Reviewer 2 Report

The article is very interesting, it presents the current state of knowledge about MXene. However, in the Aplication section the authors have omitted a fairly large and constantly evolving group of composite materials in which MXene is used as the reinforcement phase. In the literature, there are reports of the use of MXene as a reinforcement in polymer or ceramic matrix composites. I recommend reading the works of such authors as: Jing Guo, Mateusz Petrus, Tomasz Cygan, Aamir Iqbal, Mingming Fei. It is recommended to extend the manuscript with the section on composites with the addition of MXene.

Author Response

Dear Reviewer! We kindly thank you for the careful revision of our manuscript and your very important comment. The composite materials with MXene is absolutely amazing field of science and Engineering. There are some review and research articles authored by Jing Guo, Mateusz Petrus, Tomasz Cygan, Aamir Iqbal, Mingming Fei that completely covered this topic. The authors of this review are not experts in this area and do not feel confident to add a full chapter regarding the composites. We have added some information related to some papers of Jing Guo, Mateusz Petrus, Tomasz Cygan, Aamir Iqbal, Mingming Fei (reference 121-124) that significantly support information in the current review.

Reviewer 3 Report

See the attachment for my feedback

Author Response

Dear Reviewer! We kindly thank you for the careful revision of our manuscript! We correct all points you provided in the revision letter!